# MULTI-CONDITION CONFORMAL SELECTION

**Qingyang Hao[1], Wenbo Liao[1,2], Bingyi Jing[1], Hongxin Wei[1]***

[1] Department of Statistics and Data Science, Southern University of Science and Technology
[2] Department of Mathematics, The Chinese University of Hong Kong

## ABSTRACT

Selecting high-quality candidates from large-scale datasets is critically important in resource-constrained applications such as drug discovery, precision medicine, and the alignment of large language models. While conformal selection methods offer a rigorous solution with False Discovery Rate (FDR) control, their applicability is confined to single-threshold scenarios (i.e., $y > c$) and overlooks practical needs for multi-condition selection, such as conjunctive or disjunctive conditions. In this work, we propose the Multi-Condition Conformal Selection (*MCCS*) algorithm, which extends conformal selection to scenarios with multiple conditions. In particular, we introduce a novel nonconformity score with regional monotonicity for conjunctive conditions and a global Benjamini–Hochberg (BH) procedure for disjunctive conditions, thereby establishing finite-sample FDR control with theoretical guarantees. The integration of these components enables the proposed method to achieve rigorous FDR-controlled selection in various multi-condition environments. Extensive experiments validate the superiority of *MCCS* over baselines, its generalizability across diverse condition combinations, different real-world modalities, and multi-task scalability.

## 1 INTRODUCTION

In many practical scenarios with limited resources, it is essential to select a subset of candidates from a very large pool that best meet specific criteria for downstream processing. For instance, drug discovery screens compounds using properties like *logP* to avoid molecular docking pitfalls (Noor et al., 2024); medical monitoring prioritizes high-risk patients via predictive systems (Hatib et al., 2018); and large language model (LLM) research employs selective procedures for output validation (Cherian et al., 2024). Across these domains, controlling the finite-sample *false discovery rate* (FDR) is crucial for conserving resources and mitigating risks. The reliance on machine learning model predictions, due to the frequent unavailability of true test responses, necessitates quantifying uncertainty for maintaining efficiency. *cfBH* (Jin & Candes, 2023) thus formulates the selection problem as a multiple hypothesis test by employing conformal $p$-values (Bates et al., 2023), which are constructed based on conformal prediction principles (Vovk et al., 2005). This multiple testing-based framework has been validated in areas such as LLM alignment (Gui et al., 2024) and extended in further studies such as *mCS* (Bai et al., 2025b).

However, existing approaches restrict the hypothesis to a single condition of the form $y > c$, a formulation that is often insufficient in practical applications. In the concept of "drug-like" properties, compounds with medium logP are chosen by most practices (Shultz, 2019), which exemplifies the need for conjunctive conditions (i.e., $c_1 < y < c_2$). Conversely, disjunctive conditions (i.e., $y < c_1$ or $y > c_2$) are necessary in other contexts, such as early-warning systems that must activate when a target value exceeds or falls below critical thresholds (Lameski et al., 2017; Bollepalli et al., 2021). These instances highlight the necessity of selection under multi-condition settings.

In this paper, we extend the conformal selection framework to multi-condition settings, with a central goal of achieving finite-sample FDR control under both conjunctive and disjunctive conditions. We first show that naive combinations via intersection or union of single-condition selection results fail to preserve FDR control for multi-condition selection. To address this, we introduce a novel method *MCCS* equipped with rigorous theoretical guarantees. For conjunctive conditions, we de-

---

*Corresponding author (weihx@sustech.edu.cn).

sign a tailored nonconformity function that targets intervals defined by simultaneous constraints; for disjunctive conditions, we introduce a global ranking mechanism over conformal $p$-values from individual conditions, followed by application of the Benjamini–Hochberg (BH) procedure (Benjamini & Hochberg, 1995). The approach naturally generalizes to compound multi-interval targets and multivariate response settings, while maintaining finite-sample FDR guarantees.

Our method was rigorously evaluated on both simulated and real-world datasets.[1] In simulations, it demonstrated superior performance over baseline methods, as evidenced by a tighter alignment between the achieved *false discovery proportion* (FDP) and the nominal FDR. For instance, in the experiment of the univariate case under conjunctive conditions with a nominal FDR of 0.3, the FDP achieved by baselines exhibited significant deviations, either excessively high (0.3766) or too low (0.2013), while our approach achieved the closest approximation (0.2874) without exceeding the nominal level. Additionally, results on textual, visual, and multimodal tasks also validate the practicality of our method in real-world applications. The selection capability was further validated in multi-class scenarios, enabling reliable selection of individual, multiple, or similar classes.

## 2 PRELIMINARY

**Problem setup.** We let $\mathbf{x} \in \mathbb{R}^p$ represent the $p$-dimensional features, which are accessible in the entire dataset, and let $\mathbf{y} \in \mathbb{R}^1$ denote the responses [2]. In this paper, we mainly focus on the regression setting, and our work can be generalized to multi-class tasks as presented in Appendix D. The entire dataset is divided into a training dataset $\mathcal{D}_{\text{train}} = \{\mathbf{x}_i, \mathbf{y}_i\}_{i=1}^n$ and a test dataset $\mathcal{D}_{\text{test}} = \{\mathbf{x}_{n+j}\}_{j=1}^m$, where the corresponding test responses $\{\mathbf{y}_{n+j}\}_{j=1}^m$ are unobserved. The additional assumption is that samples $\{\mathbf{x}_i, \mathbf{y}_i\}_{i=1}^{n+m}$ in the dataset are drawn *i.i.d.* from an unknown distribution $\mathcal{D}_{X \times Y}$.

The selection problem can be formulated as follows: Given a union of intervals $I_{Target}$, which can be a union of multiple one-sided-unbounded or bounded open intervals $I_k$, our goal is to identify a subset $\mathcal{S} \subseteq \{1, \ldots, m\}$ from $\mathcal{D}_{\text{test}}$ such that as many test observations $j \in \mathcal{S}$ satisfy $\mathbf{y}_{n+j} \in I_{Target}$ as possible, while controlling the FDR (Benjamini & Hochberg, 1995) below a user-specified level $q$. The FDR is defined as the expectation of the FDP among all selected observations:

$$\text{FDR} = \mathbb{E}\left[\text{FDP}\right], \quad \text{FDP} = \frac{\sum_{j=1}^m \mathbf{1}\left\{j \in \mathcal{S}, \mathbf{y}_{n+j} \notin I_{Target}\right\}}{1 \vee |\mathcal{S}|}. \tag{1}$$

Since an excessively low FDR can be trivially achieved by selecting very few samples, it is also important to evaluate whether a method retains an adequate number of selections that truly meet the target criteria. This aspect is quantified by the metric power:

$$\text{Power} = \mathbb{E}\left[\frac{\sum_{j=1}^m \mathbf{1}\left\{j \in \mathcal{R}, \mathbf{y}_{n+j} \in I_{Target}\right\}}{1 \vee \sum_{j=1}^m \mathbf{1}\left\{\mathbf{y}_{n+j} \in I_{Target}\right\}}\right]. \tag{2}$$

Ideally, a practical approach should simultaneously achieve an FDP that closely approximates, yet does not exceed, the nominal FDR, while maintaining a relevant high power.

**Conformal selection.** The *cfBH* framework (Jin & Candes, 2023), building upon conformal prediction principles (Vovk et al., 2005), is designed to guarantee finite-sample false discovery rate (FDR) control in conformal selection under a single condition. It performs multiple hypothesis testing by selecting candidates whose values exceed a pre-specified threshold c. For each test sample, a hypothesis test is constructed with the following hypotheses:

$$H_{0j} : y_{n+j} \leq c \quad \text{vs.} \quad H_{1j} : y_{n+j} > c.$$

Rejection of $H_{0j}$ indicates that the $j$-th sample is selected, as it's considered satisfying the condition. With the hypothesis test, the CS framework ensures FDR control via two core steps as follows.

**First**, it employs a nonconformity score $V(\mathbf{x}, y)$ that satisfies appropriate monotonicity constraints to quantify how atypical a response is given the input. For a calibration sample with observed response, the score is computed as $V_i = V(\mathbf{x}_i, y_i)$; for a test sample with unobserved response, the score is evaluated at a predefined threshold c, yielding $\widehat{V}_{n+j} = V(\mathbf{x}_{n+j}, c)$. Then, the conformal $p$-value for

---

[1]Code available at https://github.com/hqy-new/mccs-iclr26.
[2]Later, we will extend the univariate responses to multivariate in Section 4.4.

each test sample is derived from the rank of its nonconformity score relative to the calibration set. Lower $p$-values provide stronger evidence for selection, corresponding to rejection of the null hypothesis $H_{0j}$. **Second**, the framework applies the Benjamini–Hochberg (BH) procedure (Benjamini & Hochberg, 1995; Benjamini & Yekutieli, 2001), a widely adopted method in multiple testing, to these conformal $p$-values for FDR control. The monotonicity property of the nonconformity score, combined with the BH procedure, jointly guarantees finite-sample FDR control. However, the *cfBH* framework and its existing derivatives are exclusively confined to single-condition selection, overlooking the complexities of multi-condition conformal selection. This gap will be elaborated upon in the subsequent section.

## 3 MULTI-CONDITION CONFORMAL SELECTION

In this section, we introduce the task of Multi-Condition Conformal Selection with detailed formulations, and demonstrate why the *cfBH* procedures (Jin & Candes, 2023) cannot be directly extended to this setting via simple set operations such as intersection or union.

### 3.1 CONDITIONS AND SELECTION TARGET

Single-condition CS refers to the process of comparing the response value $\mathbf{y}$ against a single threshold. In contrast, multi-condition CS involves evaluation against multiple thresholds to determine target satisfaction. The fundamental types of multi-condition CS are conjunctive and disjunctive conditions, with more complex structures emerging from their combinations. For conjunctive conditions, such as those requiring $\mathbf{y}_{n+j} > c_{k_L}$ and $\mathbf{y}_{n+j} < c_{k_R}$, the null hypothesis is defined as $H_{0j}^k : \mathbf{y}_{n+j} \in (c_{k_L}, c_{k_R})$, with the corresponding alternative hypothesis $H_{1j}^k : \mathbf{y}_{n+j} \notin (c_{k_L}, c_{k_R})$. This approach ensures that any null hypothesis, whether constructed from a single condition or a conjunction, can be mapped to a target interval $I_k$. Consequently, for disjunctive conditions, the selection target is logically extended to the union of the intervals associated with each individual hypothesis. That is, the composite hypothesis formed by combining all conditions may be expressed as follows:

$$H_{0j}^{combine} : \mathbf{y}_{n+j} \in I_{Target} \quad \text{vs.} \quad H_{1j}^{combine} : \mathbf{y}_{n+j} \notin I_{Target},$$

where $I_{Target} = \bigcup_{k=1}^K I_k$, each $I_k$ denotes a one-sided-unbounded or bounded open interval. We present an intuitive illustration for conditions in Figure 1 and the *FDR* control is applied to all $(j, k)$ pairs, where $j$ indexes samples and $k$ indexes conditions, an explanation provided in Appendix A.1.

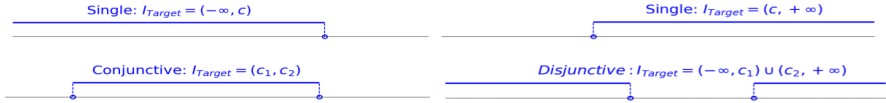

Figure 1: Illustration of single- and multi-condition.

### 3.2 CONFORMAL $p$-VALUE

The conformal $p$-value serves as the critical link between the nonconformity score and the BH procedure. When the response $\mathbf{y}_{n+j}$ is observable, we can compute the nonconformity score as $V_i = V(\mathbf{x}_i, \mathbf{y}_i)$ for $i = 1, \ldots, n + m$, where $V$ is a nonconformity function based on $\hat{\mu}$, then the oracle conformal $p$-value would be defined as:

$$p_j^* = \frac{\sum_{i=1}^n \mathbf{1}\{V_i < V_{n+j}\} + U_j \left(1 + \sum_{i=1}^n \mathbf{1}\{V_i = V_{n+j}\}\right)}{n+1} \tag{3}$$

where $U_j \sim \text{Unif}(0, 1)$ is an independent random variable for tie-breaking.

To ensure finite-sample FDR control via the BH procedure, the conformal $p$-values must satisfy conservativeness ($\mathbb{P}(p_j^* \le \alpha) \le \alpha$) (Bates et al., 2023). While this property holds for the oracle conformal $p$-value $p_j^*$, its computation is infeasible in practice since the test responses $y_{n+j}$ are unobservable. Consequently, the nonconformity score $V_{n+j} = V(\mathbf{x}_{n+j}, \mathbf{y}_{n+j})$ cannot be computed directly. Instead, we substitute it with $\widehat{V}_{n+j} = V(\mathbf{x}_{n+j}, c_k)$, where $c_k$ denotes a prespecified threshold. This leads to the conformal $p$-value conformal $p$-value definition:

$$p_j = \frac{\sum_{i=1}^n \mathbf{1}\left\{V_i < \widehat{V}_{n+j}\right\} + U_j \left(1 + \sum_{i=1}^n \mathbf{1}\left\{V_i = \widehat{V}_{n+j}\right\}\right)}{n+1}. \tag{4}$$

To ensure the conservativeness of practical $p$-values $p_j$, the nonconformity score $V(\mathbf{x}, \mathbf{y})$ must satisfy the following *regional monotonicity*:

**Definition 3.1** (Regional Monotonicity). *((Bai et al., 2025b)) A nonconformity score $V : \mathcal{X} \times \mathcal{Y} \to \mathbb{R}$ satisfies regional monotonicity if $V(\mathbf{x}, \mathbf{y}') \leq V(\mathbf{x}, \mathbf{y})$ for all $\mathbf{x} \in \mathcal{X}$, $\mathbf{y}' \in R^c$, and $y \in R$, where $R$ represents the target region.*

As shown in *mCS* (Bai et al., 2025b), the practical $p$-values $p_j$, derived from a nonconformity measure satisfying regional monotonicity, are conservative and thus ensure FDR control. Note that the conservativeness of $p_j$ here is not the usual statistical conservatism, but refers specifically to the following under $H_0$: $\mathbb{P}(p_j \leq \alpha \text{ and } j \in \mathcal{H}_0) \leq \alpha, \forall \alpha \in (0, 1)$. This property nonetheless suffices to guarantee FDR control (Bai et al., 2025b). The key challenge is thus to design nonconformity measures that preserve this monotonicity under multi-condition settings, especially for intersection conditions, which we will address in the next section.

### 3.3 Error accumulation in two-step cfBH methods

A seemingly straightforward approach for handling multiple conditions would be to directly apply the *cfBH* method (Jin & Candes, 2023) at each boundary point independently and then combine the results through simple set operations as the second step. Specifically, for a target interval $(c_1, c_2)$, one may apply *cfBH* separately at each boundary, selecting units with $Y > c_1$ and $Y < c_2$, then take their intersection (*Inter-cfBH*). For the union $(-\infty, c_1) \cup (c_2, \infty)$, one can similarly use *cfBH* at $c_1$ and $c_2$ to select $Y < c_1$ and $Y > c_2$, respectively, and combine the results via union (*Union-cfBH*).

However, these methods cannot guarantee FDR control, primarily due to error accumulation resulting from naive set operations. Such error propagation undermines their theoretical validity, necessitating a rigorously guaranteed approach for FDR control under multi-condition settings. We therefore present Corollary 3.1, supported by numerical results in Section 5.1 demonstrating the failure of both *Inter-cfBH* and *Union-cfBH*, thus providing empirical validation for the corollary.

**Corollary 3.1.** *Applying two separate cfBH procedures (Jin & Candes, 2023) for boundary points and combining results via intersection (for the target interval $(c_1, c_2)$) or union (for $(-\infty, c_1) \cup (c_2, \infty)$) does not guarantee control of the false discovery rate at a prespecified level $q$, where $c_1 < c_2$ are predefined thresholds.*

**Remark 3.1** (Explanation for Corollary 3.1). *For intersection selection $\mathcal{S} = \mathcal{S}_1 \cap \mathcal{S}_2$, a false discovery requires simultaneous errors in both procedures. For instance, a unit with $Y \leq c_1$ may be falsely selected if erroneously included by $\mathcal{S}_1$ yet retained by $\mathcal{S}_2$ (as $Y \leq c_1 < c_2$). Similarly, for $Y \geq c_2$, error necessitates incorrect exclusion by $\mathcal{S}_2$ but correct inclusion by $\mathcal{S}_1$. This multiplicative structure inflates FDR as $\mathcal{S}$ decreases, violating control guarantees. For union selection $\mathcal{S} = \mathcal{S}_1 \cup \mathcal{S}_2$, errors accumulate additively. Units with $Y_j \in [c_1, c_2]$ are falsely discovered if selected by either procedure. The numerator sums overlapping false discoveries, while correlation between $\mathcal{S}_1$ and $\mathcal{S}_2$ prevents proportional denominator growth, leading to FDR inflation. To address this, we then propose the following FDR-guaranteed method for Multi-Condition CS.*

## 4 Proposed method

In this section, we introduce the method with a finite-sample FDR guarantee for Multi-Condition CS. We begin by addressing conjunctive and disjunctive conditions, then introduce the complete *MCCS* for diverse logical combinations and extend it to multivariate response settings.

### 4.1 Addressing conjunctive conditions

Central to addressing conjunctive conditions (i.e., $c_1 < y < c_2$) is the design of the nonconformity score. For conjunctive conditions defined by the interval $I_k = (c_{k_L}, c_{k_R})$, the nonconformity score $V^k(\mathbf{x}, \mathbf{y})$ in Algorithm 1 ensures regional monotonicity by leveraging the predictor $\hat{\mu}(\mathbf{x})$ and a constant M. The score assigns

$$M - \min(\hat{\mu}(\mathbf{x}) - c_{k_L}, c_{k_R} - \hat{\mu}(\mathbf{x})) \tag{5}$$

if $\mathbf{y} \in (c_{k_L}, c_{k_R})$, and

$$\max(c_{k_L} - \hat{\mu}(\mathbf{x}), \hat{\mu}(\mathbf{x}) - c_{k_R}) \tag{6}$$

---

**Algorithm 1** Nonconformity Score for Conjunctive Conditions

1: **function** $V^k(\mathbf{x}, \mathbf{y})$               ▷ For the determined interval $I_k = (c_{k_L}, c_{k_R})$
2:     $pred \leftarrow \hat{\mu}(\mathbf{x})$,     $M \leftarrow \text{constant} > 2 \cdot \sup_x \max(|c_{k_L} - \hat{\mu}(\mathbf{x})|, |\hat{\mu}(\mathbf{x}) - c_{k_R}|)$
3:     **if** $\mathbf{y} > c_{k_L}$ **and** $\mathbf{y} < c_{k_R}$ **then return** $M - \min(pred - c_{k_L}, c_{k_R} - pred)$
4:     **else**    **return** $\max(c_{k_L} - pred, pred - c_{k_R})$
5:     **end if**
6: **end function**

---

otherwise, guaranteeing lower scores for samples within $I_k$ to promote selection. while $M$ enforces the monotonicity, and further utility of $M$ is formalized in Proposition 4.1.

**Proposition 4.1.** *Let $V^k$ be any fixed regionally monotone nonconformity score, and suppose $\{(\mathbf{x}_i, \mathbf{y}_i)\}_{i=1}^{n+m}$ are exchangeable from distribution $\mathcal{D}_{X \times Y}$. Let $(\mathbf{x}, y)$ denote a random pair drawn from $\mathcal{D}_{X \times Y}$, and define $F(v, u) = \mathbb{P}(V^k(\mathbf{x}, \mathbf{y}) < v) + u \cdot \mathbb{P}(V^k(\mathbf{x}, \mathbf{y}) = v)$ for any $v \in \mathbb{R}$ and $u \in [0, 1]$. For interval $I_{Target} = I_k = (c_{k_L}, c_{k_R})$, assuming fixed choice of $\mathbf{c}_k \in I_k$, define*

$$t^* = \sup \left\{ t \in [0, 1] : \frac{t}{\mathbb{P}(F(V^k(\mathbf{x}, \mathbf{c}_k), U) \le t)} \le q \right\}. \tag{7}$$

*Suppose there exists $t \in (t^* - \epsilon, t^*)$ such that $\frac{t}{\mathbb{P}(F(V^k(\mathbf{x}, \mathbf{c}_k), U) \le t)} \le q$ for any sufficiently small $\epsilon > 0$. Then the output $\mathcal{S}$ of Algorithm 3 (where $V^k$ adopts Algorithm 1) satisfies:*

$$\lim_{n,m \to \infty} \text{FDR} = \frac{\mathbb{P}\left(F(V^k(\mathbf{x}, \mathbf{c}_k), U) \le t^*, \mathbf{y} \in I_k^c\right)}{\mathbb{P}\left(F(V^k(\mathbf{x}, \mathbf{c}_k), U) \le t^*\right)}, \quad and \quad \lim_{n,m \to \infty} \text{Power} = \frac{\mathbb{P}\left(F(V^k(\mathbf{x}, \mathbf{c}_k), U) \le t^*, \mathbf{y} \in I_k\right)}{\mathbb{P}(\mathbf{y} \in I_k)}. \tag{8}$$

Proposition 4.1 extends Proposition 7 in *CS* (Jin & Candes, 2023) from a single condition to conjunctive conditions, establishing the exact asymptotic behavior of both the FDR and power. The proof, being highly analogous, is omitted. Here, $F(v, u)$ can be interpreted as a lower and more robust measure of evidence, while $t^*$ corresponds to the critical threshold value selected by the BH procedure. The requirement that a small neighborhood around $t^*$ satisfies the definition in (7) ensures stability in FDR control by excluding scenarios where $t^*$ exhibits oscillatory behavior.

Under the assumption that $V^k$ does not possess point masses, the meaning of $t^*$ can be interpreted more intuitively by introducing $v^*$: let $v^* = \sup \left\{ v : P\left(V^k(X, Y) \le v\right) \le t^* \right\}$, then the inequality in the definition of $t^*$ in (7) can be rewritten as:

$$\frac{t^*}{\mathbb{P}(F(V^k(\mathbf{x}, \mathbf{c}_k), U) \le t^*)} = \frac{\mathbb{P}(V^k(\mathbf{x}, \mathbf{y}) \le v^*, \mathbf{y} \in I_k^c)}{\mathbb{P}(V^k(\mathbf{x}, \mathbf{c}_k) \le v^*)} \le \frac{\mathbb{P}(V^k(\mathbf{x}, \mathbf{y}) \le v^*)}{\mathbb{P}(V^k(\mathbf{x}, \mathbf{c}_k) \le v^*)} \le q \tag{9}$$

Hence, tightening the first inequality in (9) allows the algorithm to approach the nominal FDR more closely. The large constant $M$ was introduced to achieve this objective, as detailed in Appendix A.2.

**Remark 4.1.** *Although this idea is similar to BH_clip in cfBH (Jin & Candes, 2023) in spirit, and the "fixed choice" in Proposition 4.1 corresponds to a specific criterion, as in Theorem 4.1 of mCS (Bai et al., 2025b), the criterion in Proposition 4.1 is defined by two conjunctive conditions rather than a single one in Theorem 4.1 of mCS (Bai et al., 2025b). Furthermore, Proposition 4.1 can be naturally extended to multivariate response settings with conjunctive conditions, which cannot be handled by Theorem 4.1 in (Bai et al., 2025b).*

### 4.2 Addressing disjunctive conditions

Addressing disjunctive conditions (i.e., $y < c_1$ or $y > c_2$) requires a more effective utilization of the BH procedure. To this end, we propose Algorithm 2, which implements via a global approach that aggregates all $m \times K$ $p$-values $\mathcal{P} = \{p_j^k \mid j = 1, \ldots, m; k = 1, \ldots, K\}$, sorts them ascendingly as $p_{(1)} \le \cdots \le p_{(NUM)}$ where $NUM = m \cdot K$, finds the largest index $l^*$ satisfying $p_{(l^*)} \le \frac{q \cdot l^*}{NUM}$, and constructs the selection set $\mathcal{S} = \left\{ (j, k) : p_j^k \le \frac{q \cdot l^*}{NUM} \right\}$.

This method ensures finite-sample FDR control under exchangeability (Theorem 2.6, *CS* (Jin & Candes, 2023)) by integrating all hypotheses within a unified testing framework, thus particularly effective for selecting targets spanning multiple intervals as preventing the error inflation inherent in ad hoc combinations of separate *cfBH* procedures, and providing rigorous theoretical guarantees for FDR control under disjunctive conditions.

---

**Algorithm 2** Global BH Procedure

---

1: **procedure** GLOBALBH($\{p_j^k\}, q$)     $\triangleright j = 1, \ldots, m; k = 1, \ldots, K$
2:     Collect all $m \times K$ p-values: $\mathcal{P} = \{p_j^k \mid j = 1, \ldots, m; k = 1, \ldots, K\}$
3:     Sort p-values in ascending order: $p_{(1)} \leq p_{(2)} \leq \cdots \leq p_{(NUM)}$   where $NUM = m \cdot K$
4:     Find maximum $l^*$ satisfying: $p_{(l^*)} \leq \frac{q \cdot l^*}{NUM}$
5:     Construct selection set: $\mathcal{S} = \left\{ (j, k) : p_j^k \leq \frac{q \cdot k^*}{NUM} \right\}$
6:     **return** Selection set $\mathcal{S}$     $\triangleright$ Selected pairs $(j, k)$ claiming $\mathbf{y}_{n+j} \in I_k$ with FDR $\leq q$
7: **end procedure**

---

### 4.3 ADDRESSING MORE COMBINATION CONDITIONS

Building on the method for two-condition (conjunctive or disjunctive) CS, the approach can be extended to more formats and multi-condition combinations. A one-sided unbounded interval, defined by a single constraint, represents a special case of a conjunctive condition with one infinite threshold. Specifically, the nonconformity score for a left-unbounded interval $I_k = (-\infty, c_{k_R})$ and a right-unbounded interval $I_k = (c_{k_L}, +\infty)$ is defined respectively as:

$$V^k(\mathbf{x}, \mathbf{y}) = M \cdot \mathbf{1}_{\{\mathbf{y} < c_{k_R}\}} + \hat{\mu}(\mathbf{x}), \quad V^k(\mathbf{x}, \mathbf{y}) = M \cdot \mathbf{1}_{\{\mathbf{y} > c_{k_L}\}} - \hat{\mu}(\mathbf{x}). \quad (10)$$

Therefore, we propose the complement *MCCS* in Algorithm 2, which designs regionally monotone nonconformity scores $V^k(\mathbf{x}, \mathbf{y})$ for each target interval $I_k$, computes conformal $p$-values $p_j^k$ for all test samples and intervals based on the calibration data (as in Eq. (4)), and applies the global BH procedure as in Algorithm 2 to select all pairs $(j, k)$, ensuring finite-sample FDR control.

---

**Algorithm 3** Multi-Condition Conformal Selection (*MCCS*)

---

**Input:** $\mathcal{D}_{\text{cal}} = \{(\mathbf{x}_i, \mathbf{y}_i)\}_{i=1}^n$, $\mathcal{D}_{\text{test}} = \{\mathbf{x}_{n+j}\}_{j=1}^m$, $I_{Target} = \bigcup_{k=1}^K I_k$, FDR target $q \in (0, 1)$.
1: **for** $k \leftarrow 1$ to $K$ **do**     $\triangleright$ Design nonconformity score tailored to each target interval $I_k$
2:     Define $V^k(\mathbf{x}, \mathbf{y})$ satisfying the regionally monotone.
3: **end for**
4: **for** $k \leftarrow 1$ to $K$ and $i \leftarrow 1$ to $n$ **do**     $\triangleright$ Compute nonconformity scores on calibration data
5:     Compute $V^k(\mathbf{x}_i, \mathbf{y}_i)$.
6: **end for**
7: **for** $k \leftarrow 1$ to $K$ and $j \leftarrow 1$ to $m$ **do**     $\triangleright$ Compute conformal $p$-values
8:     Compute $\widehat{V}_{n+j}^k$ , and then construct $p_j^k$ as in (4).
9: **end for**
10: $\mathcal{P} \leftarrow \text{sort}\left( \{p_j^k\}_{j=1, k=1}^{m, K} \right)$, $l^* \leftarrow \max\left\{ l : \mathcal{P}[l] \leq \frac{q \cdot l}{m \cdot K} \right\}$.     $\triangleright$ Global BH procedure

**Output:** Selection set $\mathcal{S} = \left\{ (j, k) : p_j^k \leq \frac{q \cdot l^*}{m \cdot K} \right\}$.

---

To enhance practicality, we state Corollary 4.1, which establishes that overlapping intervals within $I_{Target}$ preserve the exchangeability and conservativeness required for FDR control, as experimentally validated in Section 5.2. This property enables direct specification of multiple target intervals without explicit intersection checks. Theorem 4.1 formalizes the finite-sample FDR control guarantee of Algorithm 3, with proof provided in Appendix A.4.

**Corollary 4.1.** *Under the conditions of Theorem 4.1, intersections among target intervals $I_k$ do not affect the FDR control guaranteed by Algorithm 3.*

**Theorem 4.1.** *Suppose $V^k$ is a regionally monotone nonconformity score, and for any $j \in \{1, \ldots, m\}$, the random variables $V_1^k, \ldots, V_n^k, V_{n+j}^k$ are exchangeable conditioned on $\{\widehat{V}_{n+\ell}^k\}_{\ell \neq j}$. Then, for any $q \in (0, 1)$, the output $\mathcal{S}$ of Algorithm 3 satisfies* FDR $\leq q$.

---

### 4.4 EXTENSION TO MULTIVARIATE RESPONSE SETTINGS

Building on *mCS* (Bai et al., 2025b), Algorithm 3 extends naturally to multivariate responses. For conjunctive conditions, the target region is a bounded area $\partial R$; for disjunctive conditions, Algorithm 4

is proposed, analogous to Algorithm 1. Algorithm 2 operates on conformal p-values independently of response dimension, enabling direct multivariate application. This integration generalizes *MCCS* for multivariate settings. See Appendix A.3 for details.

---

**Algorithm 4** Nonconformity Score for Conjunctive Conditions under Multivariate Response Setting

---

1: **function** $V^k(\mathbf{x}, \mathbf{y})$ $\qquad\qquad\qquad$ ▷ For the determined region between $\partial R_{inner}$ and $\partial R_{outter}$
2: $\quad$ **pred** $\leftarrow \hat{\mu}(\mathbf{x})$, $M \leftarrow$ large constant
3: $\quad$ **if** $\mathbf{y} \notin R_{inner}$ **and** $\mathbf{y} \in R_{outter}$ **then**
4: $\quad\quad$ **return** $M - \min\{dis(\mathbf{pred}, \partial R_{inner}), dis(\mathbf{pred}, \partial R_{outter})\}$
5: $\quad$ **else** $\quad$ **return** $\max\{dis(\mathbf{pred}, \partial R_{inner}), dis(\mathbf{pred}, \partial R_{outter})\}$
6: $\quad$ **end if**
7: **end function**

---

## 5 EXPERIMENTS

We evaluated the method on both simulated and real-world data. Simulations demonstrated its superiority over baselines under conjunctive and disjunctive conditions, along with strong generalization to more complex conditional tasks. Real-world experiments confirmed its efficacy across various scenarios, including textual, visual, multimodal, and multi-class settings.

### 5.1 COMPARE WITH BASELINES

**Experiments setup.** We first perform numerical simulations using synthetically generated data. The covariates $\mathbf{x}$ are sampled from $\text{Unif}(-10, 10)^p$, where p denotes the dimension of $\mathbf{x}$. The response variable $\mathbf{y}$ is generated by $\mathbf{y} = \mu(\mathbf{x}) + \epsilon$, where $\mu$ represents the regression function and $\epsilon$ denotes the random noise. An SVM with an RBF kernel is used for model fitting. The dataset is split into training, calibration, and test sets with a ratio of 8:1:1. Different experimental settings are constructed by varying the regression function and noise distribution. The FDP averaged over 1000 independent replications serves as the observed FDR.

**Compared baselines.** We established the following baseline methods for comparison. The existing single-condition method was applied separately to each individual condition, and the resulting selection sets were combined using simple set operations: intersection (denoted as *Int*) and union (denoted as *Uni*). Bonferroni correction (Dunn, 1961) versions of these procedures are referred to as *Int-B* and *Uni-B*, respectively. Additionally, *Ind* denotes an indicator method based on whether the response variable lies within the target region (i.e., $y_{Ind} > 0$ when $\mathbf{y} \in I_{Target}$). Further details on settings and baselines are provided in Appendix B.

Table 1: Comparison with baselines of selection performance for conjunctive and disjunctive conditions under univariate and multivariate response settings (Optimal FDR control in **bold**).

(a) Conjunctive conditions

| Method | Univariate | | Multivariate | |
|---|---|---|---|---|
| | FDR | Power | FDR | Power |
| *Int* | 0.3766 | 0.9397 | 0.3531 | 0.9622 |
| *Int-B* | 0.1081 | 0.6005 | 0.2208 | 0.8549 |
| *Ind* | 0.2013 | 0.2126 | 0.1491 | 0.1179 |
| *MCCS* (Ours) | **0.2874** | 0.9756 | **0.2907** | 0.5348 |

(b) Disjunctive conditions

| Method | Univariate | | Multivariate | |
|---|---|---|---|---|
| | FDR | Power | FDR | Power |
| *Uni* | 0.3766 | 0.9720 | 0.3165 | 0.8024 |
| *Uni-B* | 0.1569 | 0.9224 | 0.1644 | 0.5765 |
| *Ind* | 0.2290 | 0.0000 | 0.2172 | 0.0044 |
| *MCCS* (Ours) | **0.2848** | 0.9515 | **0.2628** | 0.7013 |

In Table 1, we compare the observed FDR and power with baseline methods, showing results for conjunctive conditions and disjunctive conditions at a nominal FDR level of 0.3. For the multivariate response setting, we report results with a response dimension of 30. As previously discussed, applying two separate single-condition methods followed by naive set operations leads to error accumulation, consistently yielding obtained FDR values above the nominal level. In contrast, the Bonferroni correction versions show an overly conservative behavior, underscoring the need for our proposed method. The *Ind* approach also exhibits conservative FDR and lower power. In comparison, our method robustly controls the FDR at or slightly below the nominal level while maintaining a relatively high power.

## 5.2 EXPERIMENTS ON MORE COMBINATION CONDITIONS

Table 2: Configuration of $I_{Target}$ for different Tasks

| Non-Intersecting Target Intervals | Intersecting Target Intervals |
|---|---|
| **Task 1:** $(-\infty, c_{1_R}) \cup (c_{2_L}, c_{2_R})$ | **Task 4:** $(c_{1_L}, c_{1_R}) \cup (c_{2_L}, c_{2_R}) \cup (c_{3_L}, +\infty)$ |
| **Task 2:** $(c_{1_L}, c_{1_R}) \cup (c_{2_L}, +\infty)$ | **Task 5:** $(-\infty, c_{1_R}) \cup (c_{2_L}, c_{2_R}) \cup (c_{3_L}, +\infty)$ |
| **Task 3:** $(-\infty, c_{1_R}) \cup (c_{2_L}, c_{2_R}) \cup (c_{3_L}, c_{3_R})$ | **Task 6:** $(-\infty, c_{1_R}) \cup (c_{2_L}, c_{2_R}) \cup (c_{3_L}, c_{3_R}) \cup (c_{4_L}, +\infty)$ |

Table 3: Results across varying $ST$ and $Ns$ for Task5.

| | FDR | | | Power | | |
|---|---|---|---|---|---|---|
| $ST$ | $Ns$=0.1 | $Ns$=0.5 | $Ns$=0.9 | $Ns$=0.1 | $Ns$=0.5 | $Ns$=0.9 |
| 1 | 0.2971 | 0.2953 | 0.2954 | 0.9548 | 0.9522 | 0.9502 |
| 2 | 0.2910 | 0.2912 | 0.2912 | 0.8631 | 0.8630 | 0.8623 |
| 3 | 0.2970 | 0.2969 | 0.2969 | 0.8843 | 0.8841 | 0.8841 |
| 4 | 0.2966 | 0.2963 | 0.2957 | 0.9530 | 0.9526 | 0.9483 |
| 5 | 0.2913 | 0.2912 | 0.2915 | 0.8632 | 0.8628 | 0.8630 |
| 6 | 0.2970 | 0.2969 | 0.2972 | 0.8853 | 0.8853 | 0.8854 |

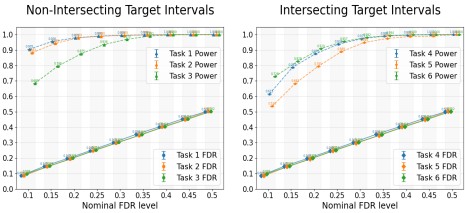

Figure 2: Results across varying $q$ for Tasks 1-6.

**Basic performance.** We designed six different tasks, as shown in Table 2, including tasks with and without intersecting intervals, each task corresponding to a distinct form of $I_{\text{Target}}$. Table 3 presents the performance of *MCCS* on Task 5 across six distinct settings (*ST1–ST6*) and three noise levels (*Ns=0.1, 0.5, 0.9*) at *q=0.3*. The results indicate a mild reduction in power as noise increases; however, the method maintains robust performance across all configurations. Figure 2 presents the performance of *MCCS* under Setting 1 across six tasks at *Ns=0.5* across varying nominal FDR levels $q$, ranging from 0.05 to 0.5 in increments of 0.05. Consistent with Corollary 4.1, intersecting target intervals does not compromise FDR control. Our method consistently maintains accurate FDR control while maintaining high power across all tasks.

Table 4: Results on large $K$ under $ST1$.

| | FDR | | | Power | | |
|---|---|---|---|---|---|---|
| $Ns$ | $K$=10 | $K$=20 | $K$=40 | $K$=10 | $K$=20 | $K$=40 |
| 0.1 | 0.2926 | 0.2886 | 0.2773 | 0.9942 | 0.9789 | 0.9069 |
| 0.5 | 0.2935 | 0.2882 | 0.2762 | 0.9951 | 0.9785 | 0.9052 |
| 0.9 | 0.2951 | 0.2888 | 0.2759 | 0.9946 | 0.9777 | 0.9038 |

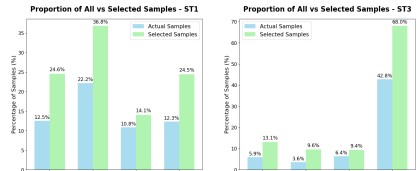

Figure 3: Interval-specific contribution.

**Analysis on large $K$.** We conduct additional experiments at *q=0.3* with a significantly larger value of interval number $K$, an order of magnitude higher than in Table 2. In these experiments, the target interval was defined as $I_{\text{Target}} = \bigcup_{k=1}^{K}(c_{k_L}, c_{k_R})$, with the Setting fixed at 1 and noise levels varied at 0.1, 0.5, and 0.9, respectively. Table 4 shows that as the number of intervals $K$ increases, statistical power decreases slightly while remaining high, and FDR control becomes more conservative while staying near the nominal level. This occurs because the BH procedure's critical value $\frac{q \cdot l}{m \cdot K}$ causes rejection regions to contract, reducing true signal detection, while the increased proportion of null hypotheses due to interval subdivision necessitates more conservative thresholds to maintain overall FDR control. These performance changes under large $K$ represent natural consequences of multiple testing adjustments, demonstrating *MCCS*'s statistical rigor.

**Analysis on interval-specific contribution.** We analyzed the selection proportions across intervals at *q=0.3* in Task 6, using Settings 1 and 3 as examples. The proportions of all test samples were compared with those of the selected samples for each target interval. Figure 3 shows a consistent distribution pattern between the *All* and *Selected* samples across intervals. This pattern indicates that when selections favor certain intervals, it reflects stronger statistical evidence (e.g., greater conformity to the target regions), arising from inherent variations in the data distribution. (Note that the sum of selection proportions exceeds *1-q* because Task 6 involves overlapping intervals.)

## 5.3 REAL DATA APPLICATIONS

Our method is applicable to diverse data types, as demonstrated in the following tasks, where the FDP from 500 independent replications serves as an estimate of the FDR.

**NLP.** In natural language processing (NLP) tasks, we identify moderately toxic content for toxicity detection, aiming to prioritize manual review for moderate-risk cases, and automatically process high- and low-risk samples. Two datasets are used: *Real-Toxicity-Prompts* (Gehman et al., 2020) (denoted as *nlp-A*) and *Pile-Toxicity-Balanced* (Gao et al., 2021; Korbak, 2023) (denoted as *nlp-B*).

**CV.** In computer vision (CV) tasks, we use the *NYU Depth Dataset V2* (Silberman et al., 2012) to select samples within a specified depth interval. This approach introduces an important spatial prior that narrows the solution space, thereby enhancing measurement accuracy and robustness in the target domain. This can be implemented by excluding training samples that fall outside the desired bounds during dataset preparation. We employ two distinct models, *ResNet* (He et al., 2016) and *ViT* (Dosovitskiy et al., 2020), for feature extraction, referred to as *cv-A* and *cv-B*, respectively.

Table 5: Results on *nlp* and *cv* Tasks.

| Task | FDR | Power |
|---|---|---|
| *nlp-A* | 0.291 | 0.575 |
| *nlp-B* | 0.289 | 0.512 |
| *cv-A* | 0.261 | 0.892 |
| *cv-B* | 0.293 | 0.814 |

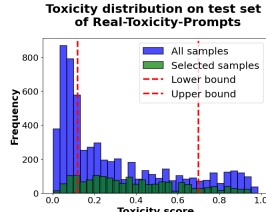 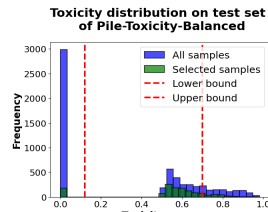

Figure 4: Visualization of *nlp*.

In Table 5, we report the observed FDR and power for both the *nlp* and *cv* tasks at a nominal FDR level of 0.3. The results demonstrate that our method maintains effective FDR control tightly below the nominal level while achieving substantial power across varying data modalities. Figure 4 visualizes the distribution of test set and selected samples in the *nlp* tasks, illustrating consistent performance across varying data structures. The selected samples remain concentrated in the target region, regardless of variations in the test set, highlighting the robustness of the approach.

**VQA.** We further evaluated our method on visual question answering (VQA) tasks incorporating both *nlp* and *cv* modalities using *VQA Dataset V2* (Lin et al., 2017). The goal is to select test samples with predicted human agreement scores within a predefined confidence interval, focusing on moderately ambiguous instances that reflect partial consensus and are most suitable for model improvement and *human–AI* collaboration. We used *BLIP* (Li et al., 2022) for confidence prediction, then predicted agreement using *Ridge* and *BayesianRidge* (denoted as *vqa-A* and *vqa-B*). Table 6 reports the FDR and power achieved on the *vqa* tasks. Figure 5 illustrates the concentration in the target area of selected samples, confirming the selection's effectiveness. These results affirm the applicability of our method to multimodal contexts and indicate its potential for scalability to even larger models. Further details are provided in Appendix C.

Table 6: Results on *vqa* Tasks.

| Task | q | FDR | Power |
|---|---|---|---|
| *vqa-A* | 0.3 | 0.263 | 0.589 |
| *vqa-A* | 0.5 | 0.483 | 0.977 |
| *vqa-B* | 0.3 | 0.285 | 0.726 |
| *vqa-B* | 0.5 | 0.477 | 0.991 |

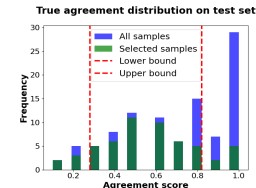

Figure 5: Visualization of *vqa*.

**Multi-class.** We extended our *MCCS* to multi-class tasks using the CIFAR-10 (Krizhevsky et al., 2009a) and CIFAR-100 (Krizhevsky et al., 2009b) datasets, enabling the selection of single, multiple, and similar classes. The achieved FDR closely aligns with the nominal FDR and maintains balanced class proportion during similar-class selection. Complete details are provided in Appendix D.

## 6 RELATED WORK

Conformal Prediction (CP) (Papadopoulos, 2008; Vovk, 2013; Vovk et al., 2005) constructs prediction sets with finite-sample coverage guarantees under the assumption of exchangeability. Despite recent progress in CP (Kiyani et al., 2024; Ge et al., 2024; Luo & Zhou, 2025; Snell & Griffiths, 2025; Kiyani et al., 2025; Zargarbashi et al., 2025; Schröder et al., 2025), they cannot be directly applied to Conformal Selection (CS) problems, which require controlling FDR in a select set rather than constructing prediction sets with coverage guarantees. This gap motivated the development of the

CS framework *cfBH* (Jin & Candes, 2023) to bridge CP principles with the selection tasks. Subsequent research has expanded this foundation into specialized extensions addressing diverse practical challenges. For instance, WCS (Jin & Candès, 2023) incorporates propensity score weighting to handle covariate shifts between calibration and test data, while mCS (Bai et al., 2025b) generalizes the framework to multivariate responses, enabling selection based on multiple outcomes. For hierarchical data structures, conformal $e$-values (Lee & Ren, 2025) have been developed to ensure FDR control in settings with grouped data. Additionally, OptCS (Bai & Jin, 2024) enables data-driven model optimization without compromising validity, and ACS (Gui et al., 2025) dynamically updates the prediction model during the selection process to enhance power. DACS (Nair et al., 2025) integrates diversity metrics to promote representative selections, and a Neyman-Pearson-inspired approach (Qin et al., 2025) achieves asymptotic optimal power under covariate shift. Applications such as LLM alignment (Gui et al., 2024) and drug discovery (Bai et al., 2025a) underscore the practical value of these advancements. However, these advances primarily focus on single-condition scenarios. Our work extends this line of research by addressing multi-condition settings while maintaining a theoretical guarantee, which is realistic in various real-world applications.

# 7 DISCUSSION OF EFFICIENCY IMPROVEMENT

To optimize the identified sorting bottleneck in the global BH procedure, based on our theoretical analysis and experimental evaluation of *MCCS*, we identify integration of the Quickselect algorithm (Hoare, 1961) enhanced the computational efficiency. This approach, termed *MCCSqck*, reduces sorting overhead by directly determining the BH threshold, avoiding full sorting. We compared *MCCS* and *MCCSqck* across varying numbers of intervals under Setting 1 and noise level 0.5.

Results are shown in Table 7, where Time refers to the time taken for 100 replications without GPU acceleration. The results demonstrate that *MCCSqck* maintains comparable FDR and Power metrics to *MCCS*, while runtime advantages become increasingly pronounced as $K$ grows, underscoring its reliability for large-scale applications. This optimization aligns with *MCCS*'s rigorous framework, offering a practical balance between efficiency and statistical robustness.

Table 7: Performance comparison of *MCCS* and *MCCSqck* (Improved efficiency in **bold**).

| $I_{Target}$ | FDR | | Power | | Time | |
|---|---|---|---|---|---|---|
| | *MCCS* | *MCCSqck* | *MCCS* | *MCCSqck* | *MCCS* | *MCCSqck* |
| K=4 | 0.3030 | 0.2971 | 0.9823 | 0.9810 | 54s | **52s**$^{\downarrow 2s}$ |
| K=10 | 0.2935 | 0.2950 | 0.9951 | 0.9951 | 151s | **124s**$^{\downarrow 27s}$ |
| K=20 | 0.2941 | 0.2945 | 0.9901 | 0.9901 | 294s | **238s**$^{\downarrow 56s}$ |
| K=40 | 0.2762 | 0.2822 | 0.9052 | 0.9610 | 560s | **469s**$^{\downarrow 91s}$ |

# 8 CONCLUSION

In this work, we introduce a theoretical extension of conformal selection to multi-condition settings, addressing an important gap in the field: while existing methods default to single-condition targets (e.g., one-sided intervals), *MCCS* systematically solves this long-overlooked problem by providing a rigorous framework for multi-condition selection with finite-sample FDR guarantees. This advancement is particularly necessary because simpler alternatives, which often rely heavily on predictors, exhibit limitations such as low power or overly conservative FDR control, as empirically demonstrated in comparisons with baseline methods. The design of *MCCS* thus emerges as a critical response to these practical shortcomings. Empirical validation across various domains confirms the practical effectiveness and adaptability of the approach, with extensions to multi-class scenarios, offering a versatile and theoretically grounded solution for selection in resource-limited scenarios.

**Limitations.** In line with standard conformal selection settings, the training and test data in our experiments were drawn from the same domain. However, domain shifts may occur in real-world applications, suggesting a direction for future research.

ACKNOWLEDGEMENTMENT

This research is supported by the Shenzhen Fundamental Research Program (Grant No. JCYJ20230807091809020). We gratefully acknowledge the support of the Center for Computational Science and Engineering at the Southern University of Science and Technology.

ETHICS STATEMENT

This paper aimed to advance the field of Machine Learning. There are many potential societal consequences of our work, none of which we feel must be specifically highlighted here.

REPRODUCIBILITY STATEMENT

To ensure the reproducibility of our findings, detailed implementation instructions for the proposed method are provided in the Appendix. The real-world datasets used in the paper are publicly available, ensuring consistent and reproducible evaluation results. Additionally, the source code is publicly available at the following URL: https://github.com/hqy-new/mccs-iclr26. These measures are intended to facilitate the verification and replication of our results by other researchers.

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

# A ALGORITHM DETAILS

## A.1 EXPLANATION FOR *FDR*

We demonstrate that applying *FDR* control to each individual interval within a multi-interval target region ensures control of the overall *FDR* for samples falling outside all target intervals. For a specified target $I_{Target}$, if selection is performed for each interval $I_k$, one can readily show:

$$
\begin{aligned}
FDR_{Neither} &= E\left[\frac{claim: \mathbf{y}_{n+j} \in I_k, \quad but: \mathbf{y}_{n+j} \in I^c_{Target}}{1 \vee |\mathcal{S}|}\right] \\
&\leq E\left[\frac{claim: \mathbf{y}_{n+j} \in I_k, \quad but: \mathbf{y}_{n+j} \in I^c_k}{1 \vee |\mathcal{S}|}\right] = FDR,
\end{aligned}
$$

Hence, maintaining *FDR* control for any individual interval $I_k$ suffices to control the overall *FDR* across all intervals. Therefore, we subsequently focus on controlling the *FDR* for each interval.

The *FDR* control is applied holistically to all $(j, k)$ pairs, where $j$ indexes test samples and $k$ indexes conditions. This means that the algorithm governs the proportion of false discoveries among all individual claims of the form "sample $j$ satisfies condition $k$.", which guarantees finite-sample *FDR* control while allowing flexible preferences for specific intervals. This granularity is unattainable with baseline methods that aggregates decisions into a sample-level binary judgment. In practice, the $(j, k)$-pair control mirrors the use of "person-times" in legal or epidemiological statistics, where each occurrence (e.g., a violation or event) is counted independently, rather than aggregating across individuals. This analogy underscores why sample-level control is insufficient: just as counting "person-times" captures the frequency of actions rather than merely the number of actors, $(j, k)$-pair control accurately reflects the multiplicity of decisions in multi-condition environments, balancing theoretical rigor with practical utility.

## A.2 EXPLANATION FOR THE LARGE CONSTANT $M$

Tightening the first inequality in (9) can be achieved by ensuring $V^k(\mathbf{x}, \mathbf{y}) \leq v^*$ so that $\mathbf{y} \notin I_k$. In an extreme case, $V^k(\mathbf{x}, \mathbf{y})$ can be defined as $-\hat{\mu}(\mathbf{x})$ when $\mathbf{y} \notin I_k$ and $+\infty$ when $\mathbf{y} \in I_k$. As a practical compromise, we define $V(x, y) = M \cdot \mathbf{1}_{\{\mathbf{y} \in I_k\}} - \min(\hat{\mu}(\mathbf{x}) - c_{k_L}, c_{k_R} - \hat{\mu}(\mathbf{x}))$, where $M > 2 \cdot \sup_x \max(|c_{k_L} - \hat{\mu}(\mathbf{x})|, |\hat{\mu}(\mathbf{x}) - c_{k_R}|)$, which leads to:

$$
\begin{aligned}
V^k(\mathbf{x}, \mathbf{y}_{\text{where } \mathbf{y} \in I_k}) &\geq 2 \cdot \sup_x \max(|c_{k_L} - \hat{\mu}(\mathbf{x})|, |\hat{\mu}(\mathbf{x}) - c_{k_R}|) - \min(\hat{\mu}(\mathbf{x}) - c_{k_L}, c_{k_R} - \hat{\mu}(\mathbf{x})) \\
&\geq 2 \cdot \sup_x \max(|c_{k_L} - \hat{\mu}(\mathbf{x})|, |\hat{\mu}(\mathbf{x}) - c_{k_R}|) - \max(|\hat{\mu}(\mathbf{x}) - c_{k_L}|, |c_{k_R} - \hat{\mu}(\mathbf{x})|) \\
&\geq \sup_x \max(|c_{k_L} - \hat{\mu}(\mathbf{x})|, |\hat{\mu}(\mathbf{x}) - c_{k_R}|) \\
&\geq V^k(\mathbf{x}, \mathbf{y}_{\text{where } \mathbf{y} \notin I_k}),
\end{aligned}
$$

This ensures that the nonconformity scores for $\mathbf{y} \in I_k$ are always larger than those for $\mathbf{y} \notin I_k$. When $q < \mathbb{P}(\mathbf{y} \notin I_k)$, it follows from the definition that $v^*$ lies below the $q$-th quantile of $V^k(\mathbf{x}, \mathbf{y})$. Consequently, the first inequality in (9) becomes an exact equality, leading to obtaining an FDR closer to the nominal level and a higher power.

For multivariate response settings, the absolute difference should be replaced by a distance metric, i.e., $M > 2 \cdot \sup_x \max(dis(\hat{\mu}(\mathbf{x}), \partial R_{inner}), dis(\hat{\mu}(\mathbf{x}), \partial R_{outter}))$, where $R$ is the target region, and $dis(\hat{\mu}(\mathbf{x}), \partial R) = \inf_{\mathbf{r} \in R^c} \|\hat{\mu}(\mathbf{x}) - \mathbf{r}\|_p$.

## A.3 DETAILS OF THE EXTENSION TO MULTIVARIATE RESPONSE SETTINGS

The *MCCS* algorithm can be naturally extended to multivariate response settings, where the target region may comprise multiple domains bounded by several constraints. For brevity, we term this generalized approach Multi-Region Conformal Selection (*MRCS*). Analogous to Algorithm 3, we provide the complete procedure for *MRCS* below, where $dis(\hat{\mu}(\mathbf{x}), \partial R) = \inf_{\mathbf{r} \in R^c} \|\hat{\mu}(\mathbf{x}) - \mathbf{r}\|_p$, and *project* denotes a mapping function used to derive the expression for $\mathbf{y}$ from the boundary equation of region $\partial R$ for computation of the nonconformity scores of test samples.

---

**Algorithm 5** Multi-Region Conformal Selection

---

**Input:** $\mathcal{D}_{\text{calib}} = \{(\mathbf{x}_i, \mathbf{y}_i)\}_{i=1}^n$, $\{\mathbf{x}_{n+j}\}_{j=1}^m$, $\{R_k\}_{k=1}^K$ (with boundary equations), $q \in (0,1)$, $\hat{\mu}(\cdot)$
(pre-trained predictor), $M > 2 \cdot \sup_x \max(dis(\hat{\mu}(\mathbf{x}), \partial R_{inner}), dis(\hat{\mu}(\mathbf{x}), \partial R_{outter})$

1: **for** $k \leftarrow 1$ to $K$ **do**                            ▷ Region-specific nonconformity scores
2:     **if** $R_k$ has inner boundary $\partial R_k^{\text{inner}}$ **then**
3:         Define $V^k(x, \mathbf{y}) = M \cdot \mathbf{1}\{\mathbf{y} \in R_k\} - dis(\hat{\mu}(x), \partial R_k^{\text{inner}})$
4:     **end if**
5:     **if** $R_k$ has outer boundary $\partial R_k^{\text{outer}}$ **then**
6:         Define $V^k(x, \mathbf{y}) = M \cdot \mathbf{1}\{\mathbf{y} \in R_k\} + dis(\hat{\mu}(x), \partial R_k^{\text{outer}})$
7:     **end if**
8:     **if** $R_k$ has both boundaries **then**
9:         Define $V^k(\hat{\mu}(\mathbf{x}), \mathbf{y}) = \begin{cases} M - \min\{dis(\hat{\mu}(\mathbf{x}), \partial R_k^{\text{inner}}), dis(\hat{\mu}(\mathbf{x}), \partial R_k^{\text{outer}})\} & \mathbf{y} \in R_k \\ -\max\{dis(\hat{\mu}(\mathbf{x}), \partial R_k^{\text{inner}}), dis(\hat{\mu}(\mathbf{x}), \partial R_k^{\text{outer}})\} & \mathbf{y} \notin R_k \end{cases}$
10:     **end if**
11: **end for**
12: **for** $k \leftarrow 1$ to $K$ and $i \leftarrow 1$ to $n$ **do**                   ▷ Compute calibration scores
13:     $V_i^k \leftarrow V^k(\mathbf{x}_i, \mathbf{y}_i)$
14: **end for**
15: **for** $k \leftarrow 1$ to $K$ and $j \leftarrow 1$ to $m$ **do**                ▷ Compute test scores and $p$-values
16:     $\mathbf{b}_k \leftarrow project(\partial R_k)$, $\widehat{V}_j^k \leftarrow V_k(\mathbf{x}_j, \mathbf{b}_k)$
17:     Then construct $p_j^k$ as in (4)
18: **end for**
19: $\mathcal{P} \leftarrow \text{sort}\left(\{p_j^k\}_{j=1,k=1}^{m,K}\right)$, $l^* \leftarrow \max\left\{l : \mathcal{P}[l] \leq \dfrac{q \cdot l}{m \cdot K}\right\}$.                ▷ Global BH procedure

**Output:** Selection set $\mathcal{S} = \left\{(j,k) : p_j^k \leq \dfrac{q \cdot l^*}{m \cdot K}\right\}$.

---

### A.4 PROOF OF THEOREM 4.1

**Exchangeability Condition.** Given the *i.i.d.* data and fixed model, for any fixed $(j, k)$, the random variables $V_1^k, \dots, V_n^k, V_{n+j}^k$ are exchangeable. More precisely, conditional on the estimated test scores $\{\widehat{V}_{n+\ell}^h : \ell \neq j, h = 1, \dots, K\}$, these scores are exchangeable. This is because the calibration scores are based on *i.i.d.* data, and the true test score $V_{n+j}^k$ comes from the same distribution. Conditioning on the estimated scores of other test points does not affect exchangeability, as these estimates are based on independent test points and a fixed model.

Consequently, for any null hypothesis $(j, k) \in \mathcal{H}_0$, the oracle $p$-value $p_j^{k*}$ is superuniform conditional on the other $p$-values. That is, for any $t \in [0,1]$, $\mathbb{P}(p_j^{k*} \leq t \mid \mathcal{F}_{-j,-k}) \leq t$, where $\mathcal{F}_{-j,-k}$ is the $\sigma$-algebra generated by all other $p$-values, i.e., $\{p_l^h : (l, h) \neq (j, k)\}$.

**Properties of the BH Procedure.** The BH procedure has key properties used in the proof. Let $R = |\mathcal{S}|$, and $R_{-j,-k} = |\mathcal{S}_{-j,-k}|$, where $\mathcal{S}_{-j,-k}$ is the selection set obtained by applying the BH procedure to all $p$-values except $p_j^k$. Then, $R \geq R_{-j,-k}$ because adding a $p$-value cannot decrease the number of rejections. Moreover, if $p_j^k > \frac{q \cdot R_{-j,-k}}{NUM}$, then adding $p_j^k$ does not change the rejection count, so $R = R_{-j,-k}$. These properties are standard and are utilized in the proof.

***Proof of Theorem 4.1.*** We now prove that under the exchangeability condition and regional monotonicity, the *MCCS* algorithm controls FDR $\leq q$.

Begin by decomposing the FDR:

$$\text{FDR} = \mathbb{E}\left[\frac{|\mathcal{S} \cap \mathcal{H}_0|}{|\mathcal{S}| \vee 1}\right] = \sum_{(j,k) \in \mathcal{H}_0} \mathbb{E}\left[\frac{\mathbf{1}\{(j,k) \in \mathcal{S}\}}{|\mathcal{S}|}\right],$$

since for $(j, k) \in \mathcal{H}_0$, $\mathbf{1}\{(j,k) \in \mathcal{S} \text{ and } \mathbf{y}_{n+j} \notin I_k\} = \mathbf{1}\{(j,k) \in \mathcal{S}\}$.

Let $R = |\mathcal{S}|$, $NUM = m \cdot K$. For any $(j, k) \in \mathcal{H}_0$, the event $(j, k) \in \mathcal{S}$ is equivalent to $p_j^k \leq \frac{q \cdot R}{NUM}$ by the definition of the BH procedure. Thus,

$$\mathbb{E}\left[\frac{\mathbf{1}\{(j, k) \in \mathcal{S}\}}{R}\right] = \mathbb{E}\left[\frac{\mathbf{1}\{p_j^k \leq \frac{q \cdot R}{NUM}\}}{R}\right].$$

Since $\mathbf{y}_{n+j} \notin I_k$, regional monotonicity implies $p_j^{k*} \leq p_j^k$, so

$$\mathbf{1}\{p_j^k \leq \frac{q \cdot R}{NUM}\} \leq \mathbf{1}\{p_j^{k*} \leq \frac{q \cdot R}{NUM}\},$$

and hence

$$\mathbb{E}\left[\frac{\mathbf{1}\{p_j^k \leq \frac{q \cdot R}{NUM}\}}{R}\right] \leq \mathbb{E}\left[\frac{\mathbf{1}\{p_j^{k*} \leq \frac{q \cdot R}{NUM}\}}{R}\right].$$

Now, introduce conditional expectation. Let $\mathcal{F}_{-j,-k}$ be the $\sigma$-algebra generated by all $p$-values except $p_j^k$. Let $R_{-j,-k}$ be the number of rejections when applying the BH procedure to all p-values except $p_j^k$. By the properties of the BH procedure, $R \geq R_{-j,-k}$, and if $p_j^k > \frac{q \cdot R_{-j,-k}}{NUM}$, then $R = R_{-j,-k}$. Therefore,

$$\frac{\mathbf{1}\{p_j^k \leq \frac{q \cdot R}{NUM}\}}{R} \leq \frac{\mathbf{1}\{p_j^k \leq \frac{q \cdot R_{-j,-k}}{NUM}\}}{R_{-j,-k}}.$$

This holds because if $p_j^k \leq \frac{q \cdot R_{-j,-k}}{NUM}$, then since $R \geq R_{-j,-k}$, we have $\frac{q \cdot R}{NUM} \geq \frac{q \cdot R_{-j,-k}}{NUM}$, so $p_j^k \leq \frac{q \cdot R}{NUM}$, and $\frac{1}{R} \leq \frac{1}{R_{-j,-k}}$ (as $R \geq R_{-j,-k} > 0$), making the left side less than or equal to the right side. If $p_j^k > \frac{q \cdot R_{-j,-k}}{NUM}$, the right side is zero, and $R = R_{-j,-k}$ implies the left side is also zero. Thus,

$$\mathbb{E}\left[\frac{\mathbf{1}\{p_j^k \leq \frac{q \cdot R}{NUM}\}}{R}\right] \leq \mathbb{E}\left[\frac{\mathbf{1}\{p_j^k \leq \frac{q \cdot R_{-j,-k}}{NUM}\}}{R_{-j,-k}}\right].$$

Since $p_j^{k*} \leq p_j^k$, we have

$$\mathbf{1}\{p_j^k \leq \frac{q \cdot R_{-j,-k}}{NUM}\} \leq \mathbf{1}\{p_j^{k*} \leq \frac{q \cdot R_{-j,-k}}{NUM}\},$$

so

$$\mathbb{E}\left[\frac{\mathbf{1}\{p_j^k \leq \frac{q \cdot R_{-j,-k}}{NUM}\}}{R_{-j,-k}}\right] \leq \mathbb{E}\left[\frac{\mathbf{1}\{p_j^{k*} \leq \frac{q \cdot R_{-j,-k}}{NUM}\}}{R_{-j,-k}}\right].$$

Next, apply superuniformity. Conditional on $\mathcal{F}_{-j,-k}$, $R_{-j,-k}$ is fixed, and $p_j^{k*}$ is superuniform:

$$\mathbb{P}(p_j^{k*} \leq t \mid \mathcal{F}_{-j,-k}) \leq t \quad \text{for any } t \in [0, 1].$$

Therefore,

$$\mathbb{E}\left[\frac{\mathbf{1}\{p_j^{k*} \leq \frac{q \cdot R_{-j,-k}}{NUM}\}}{R_{-j,-k}} \mid \mathcal{F}_{-j,-k}\right] \leq \frac{q \cdot R_{-j,-k}/NUM}{R_{-j,-k}} = \frac{q}{NUM}.$$

Taking expectation,

$$\mathbb{E}\left[\frac{\mathbf{1}\{p_j^{k*} \leq \frac{q \cdot R_{-j,-k}}{NUM}\}}{R_{-j,-k}}\right] \leq \frac{q}{NUM}.$$

Combining the inequalities,

$$\mathbb{E}\left[\frac{\mathbf{1}\{p_j^k \leq \frac{q \cdot R}{NUM}\}}{R}\right] \leq \frac{q}{NUM}.$$

Finally, sum over all null hypotheses:

$$\text{FDR} = \sum_{(j,k) \in \mathcal{H}_0} \mathbb{E}\left[\frac{\mathbf{1}\{(j, k) \in \mathcal{S}\}}{R}\right] \leq \sum_{(j,k) \in \mathcal{H}_0} \frac{q}{NUM} = q \cdot \frac{|\mathcal{H}_0|}{NUM} \leq q,$$

since $|\mathcal{H}_0| \leq NUM$.

$\square$

## A.5 Additional Explanation

In Algorithm 2 and Algorithm 5, when $M$ is large, the subtracted term becomes negligible and can be omitted. Similarly, calibration samples outside the target selection have minimal influence and may also be excluded, thereby conserving computational resources.

# B Details of Numerical Experiments

## B.1 Data Generating Processes for Univariate Responses Settings

The synthetic datasets are systematically constructed to evaluate methodological performance under controlled conditions with varying degrees of complexity, encompassing a spectrum from linear to highly nonlinear relationships paired with different noise characteristics. Settings 1-3 are designed with distinct functional structures: Setting 1 embodies a linear data-generating mechanism, Setting 2 introduces moderate nonlinearity through a combination of exponential and trigonometric components, and Setting 3 further increases model intricacy with multi-feature nonlinear interactions.

To rigorously assess robustness under different stochastic environments, Settings 4-6 adopt the same functional relationships as Settings 1-3, respectively, but incorporate Laplacian noise instead of Gaussian noise, thereby introducing heavy-tailed disturbances that present additional modeling challenges. Across all configurations, independent noise, Gaussian in Settings 1-3 and Laplacian in Settings 4-6, is added with zero mean and comparable variance, which corresponds to the noise level varying in Section 5.2, to ensure equitable comparison of noise resilience. Detailed mathematical formulations for each setting are explicitly provided in Table 8, enabling a transparent and reproducible evaluation of model behavior under diverse data-generating processes and noise regimes.

Table 8: True regression functions and noise distributions

| Setting | $\mu(\mathbf{x})$ | $\epsilon$ |
|---------|-------------------|------------|
| 1 | $\sum_{j=1}^{p} \beta_j x_j$ with $\beta_j \sim \mathcal{N}(0,1)$ | $\mathcal{N}(0, \sigma^2)$ |
| 2 | $\exp(x_1) + \sin(\pi \cdot x_2)$ | $\mathcal{N}(0, \sigma^2)$ |
| 3 | $\exp(x_1) + \sin(\pi \cdot x_2) + \exp(x_3) + \sin(\pi \cdot x_4) + \exp(x_5) + \sin(\pi \cdot x_6)$ | $\mathcal{N}(0, \sigma^2)$ |
| 4 | Same as Setting 1 | $\mathcal{L}(0, \lambda)$ |
| 5 | Same as Setting 2 | $\mathcal{L}(0, \lambda)$ |
| 6 | Same as Setting 3 | $\mathcal{L}(0, \lambda)$ |

## B.2 Details of the Baseline Method *Ind*

The method provides a unified algebraic approach to transform the geometric condition of whether a point lies within one or multiple target regions into an equivalent analytical condition expressed via the sign of a scalar-valued function.

In the univariate case, the target set consists of intervals, such as $(a, b) \cup (c, d)$. A polynomial function $f(t)$ is constructed as the negative product of shifted linear factors:

$$f(t) = -(t - a)(b - t)(t - c)(d - t).$$

A point Y belongs to the union of the intervals if and only if $f(Y) > 0$. This criterion converts the set-membership condition into an inequality constraint amenable to algorithmic evaluation.

The multivariate extension, take 2-dimension as an instance, generalizes this idea to multiple curved regions—each defined by a quadratic inequality of the form $(x - h_i)^2 + (y - k_i)^2 < r_i^2$. The overall target region is defined by Boolean combinations of such inequalities. The function $f(u, v)$ is constructed as the negative product of the left-hand sides of these quadratic expressions:

$$f(u, v) = -\prod_{i=1}^{m} \left[ (u - h_i)^2 + (v - k_i)^2 - r_i^2 \right].$$

Then, a point $Y = (Y_1, Y_2)$ lies within the target region if and only if $f(Y_1, Y_2) > 0$. This construction effectively encodes the geometry of multiple intersecting circles into a single differentiable function whose sign indicates membership.

This method offers a computationally tractable approach to transform geometric constraints into algebraic inequalities applicable to both univariate and multivariate response settings. However, its performance is often suboptimal.

### B.3 DETAILS OF TASKS IN SECTION 5.2

We constructed six distinct tasks in Section 5.2, encompassing both scenarios with intersecting target intervals and those without overlaps. To detail the experimental parameters and offer a schematic overview, Figure 6 is provided below.

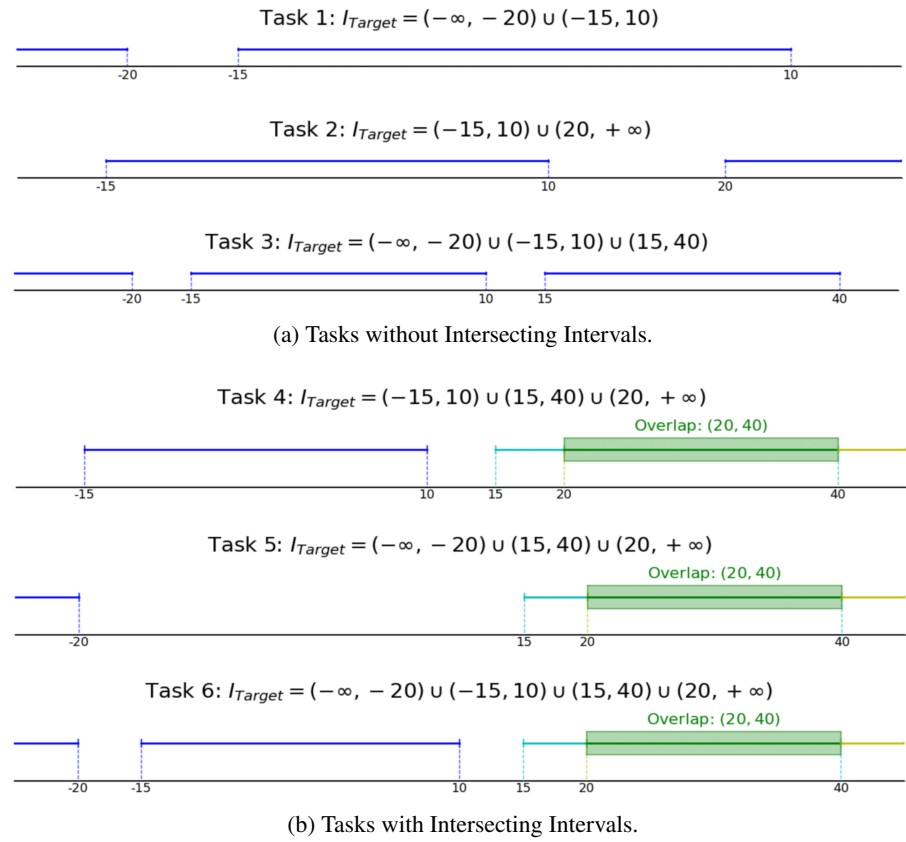

Figure 6: An Intuitive Comparison of Different Tasks.

### B.4 SUPERIORITY OF MCCS OVER $Ind$ ON INTERVAL-SPECIFIC PREFERENCE

MCCS considers the $p$-value of each interval, thereby not only improving our overall performance but also allowing us to control the selection ratio for each interval, which $Ind$-like methods, which treat all conditions uniformly, cannot do. We hereby theoretically demonstrate that this fine-grained control will maintain FDR control.

**1. Problem setup**

In MCCS, we consider $m$ test samples and $K$ target intervals, resulting in a total of $N = m \times K$ hypothesis tests. Each hypothesis $H_{jk}$ corresponds to test sample $j$ and target interval $I_k$.

We introduce predefined and data-independent weights $w_{jk} > 0$ to reflect the importance or prior information of different hypotheses, and define the weighted $p$-value as:

$p_{jk}^{\text{weighted}} = \frac{p_{jk}}{w_{jk}}$

## 2. Conditions Required for Weights

To ensure FDR control, the weights must satisfy the following conditions:

1. Non-negativity: $w_{jk} > 0$ for all $j$, $k$.

2. Normalization: $\frac{1}{N} \sum_{j=1}^{m} \sum_{k=1}^{K} w_{jk} = 1$, i.e., $\sum_{j,k} w_{jk} = N$.

3. Null Hypothesis Weight Constraint: $\sum_{(j,k) \in \mathcal{H}_0} w_{jk} \leq N$, where $\mathcal{H}_0$ is the set of null hypotheses (hypotheses where the true condition does not hold).

Note: Condition *3* is equivalent to requiring that the average weight of null hypotheses does not exceed *1*, but this is automatically satisfied because:

- From Condition *2*, the average weight of all hypotheses is 1.
- The sum of weights for null hypotheses $\leq$ the total weight sum $= N$.
- Thus, $\frac{\sum_{\mathcal{H}_0} w_{jk}}{\mathcal{H}_0} \leq \frac{N}{\mathcal{H}_0} \leq \frac{N}{1} = N$ (though this is not strictly needed).

In practice, Condition *3* is automatically satisfied because the sum of weights for null hypotheses cannot exceed the total weight sum $N$.

## 3. Weighted BH Algorithm

The algorithm steps are as follows:

1. Compute the weighted $p$-values: $p_{jk}^{\text{weighted}} = p_{jk}/w_{jk}$.

2. Sort the weighted $p$-values in ascending order: $p_{(1)}^{\text{weighted}} \leq p_{(2)}^{\text{weighted}} \leq \cdots \leq p_{(N)}^{\text{weighted}}$.

3. Find the largest index $l$ such that: $p_{(l)}^{\text{weighted}} \leq \frac{q \cdot l}{N}$.

4. Reject the corresponding $l$ hypotheses.

## 4. Proof Process

### 4.1. Notation Definitions

- $\mathcal{H}_0$: Set of null hypotheses $(y_{n+j} \notin I_k)$, $R_{jk} = \mathbb{I}(H_{jk} \text{ is rejected})$.
- $\mathcal{S} = \{(j,k) : R_{jk} = 1\}$, $R = \mathcal{S}$, FDR $= \mathbb{E}\left[\frac{V}{R \vee 1}\right]$.

### 4.2. Proof

FDR $= \sum_{(j,k) \in \mathcal{H}_0} \mathbb{E}\left[\frac{\mathbb{I}(H_{jk} \text{ is rejected})}{R \vee 1}\right]$.

Define the event $E_{jk} = \{p_{jk}^{\text{weighted}} \leq \frac{q \cdot R}{N}\}$. Then: FDR $= \sum_{(j,k) \in \mathcal{H}_0} \mathbb{E}\left[\frac{\mathbb{I}(E_{jk})}{R \vee 1}\right]$.

Let $\mathcal{F}_{-jk}$ be the $\sigma$-algebra generated by all $p$-values and weights except $p_{jk}$. Define $R_{-jk}$ as the number of rejections obtained by applying the weighted BH procedure after excluding $H_{jk}$.

From the properties of the BH procedure:

- $R \geq R_{-jk}$.
- If $p_{jk}^{\text{weighted}} > \frac{q \cdot R_{-jk}}{N}$, then $R = R_{-jk}$.

Thus: $\frac{\mathbb{I}(E_{jk})}{R \vee 1} \leq \frac{\mathbb{I}\left(p_{jk}^{\text{weighted}} \leq \frac{q \cdot R_{-jk}}{N}\right)}{R_{-jk} \vee 1}$.

Take the conditional expectation: $\mathbb{E}\left[\frac{\mathbb{I}(E_{jk})}{R \vee 1} \mid \mathcal{F}_{-jk}\right] \leq \mathbb{E}\left[\frac{\mathbb{I}\left(p_{jk}^{\text{weighted}} \leq \frac{q \cdot R_{-jk}}{N}\right)}{R_{-jk} \vee 1} \mid \mathcal{F}_{-jk}\right]$.

Given $\mathcal{F}_{-jk}$, $R_{-jk}$ is fixed. The event $p_{jk}^{\text{weighted}} \leq \frac{q \cdot R_{-jk}}{N}$ is equivalent to $p_{jk} \leq \frac{q \cdot R_{-jk} \cdot w_{jk}}{N}$.

By the conservativeness of the $p$-values (for $H_{jk} \in \mathcal{H}_0$): $\mathbb{P}\left(p_{jk} \leq \alpha | \mathcal{F}_{-jk}\right) \leq \alpha \quad \forall \alpha \in [0, 1]$.

Set $\alpha = \frac{q \cdot R_{-jk} \cdot w_{jk}}{N}$, yielding: $\mathbb{E}\left[\mathbb{I}\left(p_{jk}^{\text{weighted}} \leq \frac{q \cdot R_{-jk}}{N}\right) \middle| \mathcal{F}_{-jk}\right] \leq \frac{q \cdot R_{-jk} \cdot w_{jk}}{N}$.

Therefore: $\mathbb{E}\left[\frac{\mathbb{I}\left(p_{jk}^{\text{weighted}} \leq \frac{q \cdot R_{-jk}}{N}\right)}{R_{-jk} \vee 1} \middle| \mathcal{F}_{-jk}\right] \leq \frac{q \cdot w_{jk}}{N}$ when $R_{-jk} \geq 1$. When $R_{-jk} = 0$, both sides are zero.

Take the full expectation: $\mathbb{E}\left[\frac{\mathbb{I}(E_{jk})}{R \vee 1}\right] \leq \frac{q \cdot w_{jk}}{N}$.

Sum over all null hypotheses: $\text{FDR} \leq \sum_{(j,k) \in \mathcal{H}_0} \frac{q \cdot w_{jk}}{N} = q \cdot \frac{\sum_{(j,k) \in \mathcal{H}_0} w_{jk}}{N}$.

From the weight condition $\sum_{(j,k) \in \mathcal{H}_0} w_{jk} \leq \sum_{j,k} w_{jk} = N$.

We obtain: $\text{FDR} \leq q \cdot \frac{\sum_{\mathcal{H}_0} w_{jk}}{N} \leq q \cdot \frac{N}{N} = q$.

### 4.3. Boundary Cases

- When $R = 0$, $FDR = 0$, satisfying the control condition.
- When all weights are equal ($w_{jk} = 1$), the method reduces to the standard BH procedure.

## 5. Experimental results

If no weights are set, the proportion of samples in each interval of the selection set is consistent with the trend of the overall sample distribution. If needed, weights can be integrated to control the proportion of each interval in the selection set. We take the results of Task 6 under Setting 3 as an example in Figure 7.

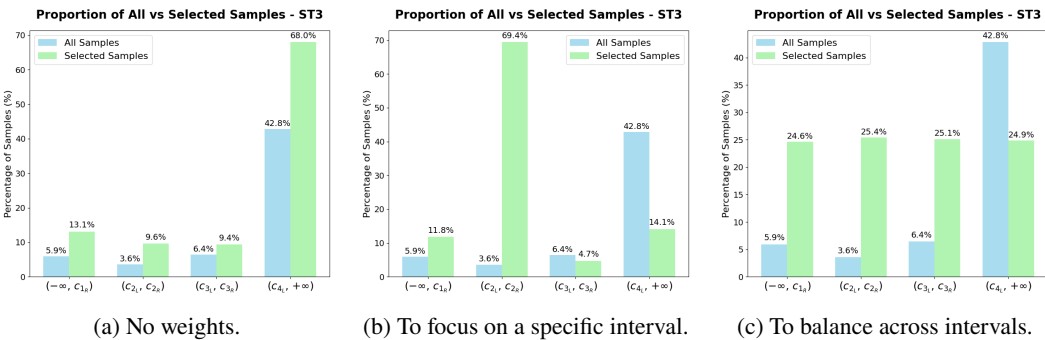

(a) No weights.      (b) To focus on a specific interval.      (c) To balance across intervals.

Figure 7: Results of different interval preference settings for Task 6 under Setting 3.

### B.5 EXPERIMENTS ON MORE BASELINES

To strengthen the claims, we evaluated another two baseline methods: the Benjamini-Yekutieli (BY) procedure and the Prediction Interval (PI) method, which provide valuable contrasts to MCCS.

**Benjamini-Yekutieli (BY) Procedure.** The *BY* procedure is a conservative adaptation of the Benjamini-Hochberg (BH) method that accounts for potential dependencies among $p$-values by incorporating a harmonic number correction. Specifically, it adjusts the rejection threshold to $\frac{q \cdot l}{m \cdot K \cdot c(m \cdot K)}$, where $c(m) = \sum_{i=1}^{m} 1/i$ accounts for dependence structures.

**Prediction Interval (PI) Method.** The *PI* method relies on parametric regression-based prediction intervals, such as those derived under normal error assumptions, to select samples whose intervals overlap with the target region. For instance, for a test sample $x_j$, the method computes a prediction interval $[L_j, U_j]$ at the $1 - q$ level and selects samples satisfying $[L_j, U_j] \cap I_{\text{Target}} \neq \emptyset$.

We compared the performance of these two baselines and MCCS on Tasks 1-6 with a Setting of $1$ and a noise level of $0.5$. The results are shown in Table 9, with the best results (closest to the nominal FDR and highest power) highlighted in bold.

Table 9: Comparison with More Baselines on Task 1-6 under Setting 1 (Best results in **bold**).

| Task | FDR | | | Power | | |
|------|-----|-----|------|-------|-----|------|
| | *BY* | *PI* | *MCCS* | *BY* | *PI* | *MCCS* |
| *1* | 0.0353 | 0.5039 | **0.2987** | 0.6242 | 0.8906 | **0.9954** |
| *2* | 0.0369 | 0.4057 | **0.2995** | 0.6522 | 0.9848 | **0.9952** |
| *3* | 0.0335 | 0.4436 | **0.3037** | 0.2181 | 0.9652 | **0.9697** |
| *4* | 0.0375 | 0.4106 | **0.3009** | 0.2452 | **0.9942** | 0.9732 |
| *5* | 0.0243 | 0.3562 | **0.2965** | 0.1428 | **0.9641** | 0.9487 |
| *6* | 0.0239 | 0.4021 | **0.3030** | 0.2240 | 0.9692 | **0.9823** |

While the BY procedure theoretically ensures FDR control under broader conditions, our experimental results show that it often leads to excessively conservative FDR estimates (i.e., FDR significantly below the nominal level) and substantially reduced power. This conservatism stems from its stringent threshold adjustment, which diminishes the ability to detect true positives, particularly in multi-condition settings where the number of hypotheses ($m \times K$) is large. In contrast, the PI method relies on traditional regression-based prediction intervals, such as those assuming normal error distributions, to select samples whose intervals overlap with the target region. This approach is computationally straightforward and can achieve high power by leveraging model-based uncertainty directly. However, our experiments reveal that PI methods frequently exceed the FDR control target, especially when model assumptions are violated or in finite-sample scenarios. This lack of robustness highlights the limitations of parametric methods in complex, real-world applications where distributional assumptions may not hold.

Only our MCCS demonstrates a balanced performance, maintaining FDR control at or near the nominal level while preserving high power across diverse conditions. This is achieved through its innovative use of $p$-values with regionally monotone nonconformity scores and the global BH procedure, which adaptively handles dependencies and multi-condition structures without requiring stringent assumptions. The experiments confirm that MCCS outperforms both BY and PI methods: it avoids the excessive conservatism of BY and the FDR inflation of PI, thereby offering a robust and practical solution for selective inference. Incorporating these baseline comparisons not only enriches the experimental section but also underscores the unique value of MCCS. By demonstrating superiority over methods that represent different approaches, we provide comprehensive evidence for MCCS's effectiveness. This expanded analysis would solidify the claims that MCCS achieves an optimal trade-off between FDR control and power in multi-condition environments.

### B.6 DETAILS OF MULTIVARIATE RESPONSE SETTINGS

**Data Generation.** We use a dataset generation function to produce multivariate regression data with systematically controlled feature-response relationships and correlated noise structures. It enables the evaluation of multidimensional regression methods under varying conditions of dimensionality and noise correlation. For each observation, the predictor variables $\mathbf{x}$ are uniformly sampled from the interval [-1, 1], forming an $n \times d$ design matrix where $d$ denotes the feature dimension (default 5). The response matrix $\mathbf{y} \in \mathbb{R}^{n \times k}$ is constructed through a structured linear combination: each dimension $Y_i$ is generated as $y_i = 2 \cdot x_j - 0.5 \cdot x_{j+1} + x_{j+2} + 1.5$, with $j = i \mod d$. This results in a rotating feature selection pattern across response dimensions, establishing a consistent yet varied linear relationship between predictors and responses.

To emulate realistic correlated noise, multivariate Gaussian noise is introduced via a covariance matrix $\Sigma$ with a specific structure: all diagonal elements are equal, and all off-diagonal elements are also equal, reflecting a homogeneous variance and uniform correlation across all response dimensions. Here, $\sigma$ controls the overall noise level and $\rho$ sets the magnitude of correlation between

any two dimensions. This setup allows fine-grained control over both the signal-to-noise ratio and the inter-dimensional correlation pattern. The data generation strategy supports rigorous evaluation of regression methods across several key aspects: the number of response variables $k$, the noise intensity $\sigma$, the correlation strength $\rho$, and the predictor dimension $d$, providing a transparent and reproducible framework for assessing model performance under controlled settings.

**Task Settings.** We define the conjunctive conditions for multivariate responses as a ring-shaped region centered at $(2, 2, \ldots, 2)$ across all dimensions, forming a symmetric reference frame. This annular region is bounded by two concentric hyperspheres whose radii are scaled proportionally from a predefined dimension-specific base value. A point is considered within the region if its Euclidean distance to the center lies between 0.6 and 1.0 times the reference radius, thus creating a well-defined annular subspace for conjunctive conditions under multivariate settings.

We define the disjunctive conditions for multivariate responses as a multi-sphere region, constructed from several hyperspheres, each with distinct centers and radii. The primary sphere is centered at $(2, 2, \ldots, 2)$, while subsequent spheres are displaced via controlled random offsets uniformly sampled from $[-0.5, 0.5]^d$, with offsets growing proportionally to the sphere index. The radii increase gradually from 80% of the base value, thus forming a cluster of intersecting hyperspheres with systematically varying positions and sizes that produce complex overlapping regions suitable for evaluating multivariate methods for disjunctive conditions.

Based on the aforementioned framework, we performed a series of simulation studies with response dimensions set to $d = 10, 30$, and $50$. The corresponding radii used in each configuration are summarized in Table 10. In these configurations, the task involving the union of conditions is referred to as *Spherical Shell*; we provide the outer sphere radius, while the inner sphere radius is set to 0.6 times the outer radius, as described earlier. For the union of multiple spherical conditions, experiments were conducted with varying numbers $num_{sp} = 2, 4$, and $8$ of spheres, and the reference sphere radius is reported for each case.

Table 10: Radius Settings for Multivariate Tasks with Varying Dimensionality of $\mathbf{y}$.

| Task | $d = 10$ | $d = 30$ | $d = 50$ |
|---|---|---|---|
| *Spherical Shell* | 5.1 | 8.1 | 10.5 |
| 2 *Spheres* | 4.4 | 7.1 | 8.9 |
| 4 *Spheres* | 4.4 | 7.2 | 9.5 |
| 8 *Spheres* | 4.9 | 8.2 | 10.5 |

**More Results.** Under the experimental framework outlined above, which aligns with the univariate configuration, we employed an SVR model with an RBF kernel as the regression method. The observed FDR was estimated by averaging the FDP over 100 repeated experimental trials. In this section, we present a more comprehensive set of results than those discussed in Section 5.

For the conjunctive conditions, we compared our approach with a baseline method, denoted as *Int*, which directly applies the intersection of two single-condition selection procedures. The experimental outcomes, summarized in Table 11, reveal that our method consistently achieves effective FDR control across varying response dimensions. In contrast, the *Int* baseline consistently produced FDR values exceeding the nominal level, underscoring the deficiency in its theoretical guarantees. Although the *Int* method occasionally attained higher statistical power than our approach, this apparent advantage came at the expense of a loss in FDR control. It is important to note that pursuing higher power alone, for instance, by selecting all samples, is trivial and statistically uninformative. Hence, the ability to maintain high power under strict FDR constraints represents a meaningful and advantageous property of our method.

For the disjunctive conditions, we compared our approach with a baseline method denoted as *Uni*, which simply applies the union of two single-condition selection procedures. Experiments were conducted across varying response dimensions, each involving target sets composed of different numbers of spheres. The results, summarized in Table 12, demonstrate that our method consistently maintains effective FDR control with relatively high power under all dimensional and number of spheres settings. In contrast, the *Uni* baseline consistently produced FDR values exceeding the

Table 11: Comparison with Baseline of Selection Performance for Conjunctive Conditions under Multivariate Response Settings (Optimal FDR control in **bold**).

| Dimension of $\mathbf{y}$ | FDR | | Power | |
|---|---|---|---|---|
| | *Int* | *MCCS* | *Int* | *MCCS* |
| *d=10* | 0.3534 | **0.2998** | 0.9696 | 0.8878 |
| *d=30* | 0.4195 | **0.2940** | 0.9761 | 0.4737 |
| *d=50* | 0.3469 | **0.2931** | 0.9787 | 0.6338 |

Table 12: Comparison with Baseline of Selection Performance for Disjunctive Conditions under Multivariate Response Settings (Optimal FDR control in **bold**).

(a) $num_{sp} = 2$

| Dimension of $\mathbf{y}$ | FDR | | Power | |
|---|---|---|---|---|
| | *Uni* | *MCCS* | *Uni* | *MCCS* |
| *d=10* | 0.3079 | **0.2679** | 0.7641 | 0.6977 |
| *d=30* | 0.3164 | **0.2788** | 0.7611 | 0.7718 |
| *d=50* | 0.3133 | **0.2833** | 0.7484 | 0.7794 |

(b) $num_{sp} = 4$

| Dimension of $\mathbf{y}$ | FDR | | Power | |
|---|---|---|---|---|
| | *Uni* | *MCCS* | *Uni* | *MCCS* |
| *d=10* | 0.3125 | **0.2683** | 0.7584 | 0.6759 |
| *d=30* | 0.3178 | **0.2822** | 0.7877 | 0.7236 |
| *d=50* | 0.3094 | **0.2751** | 0.8512 | 0.7905 |

(c) $num_{sp} = 8$

| Dimension of $\mathbf{y}$ | FDR | | Power | |
|---|---|---|---|---|
| | *Uni* | *MCCS* | *Uni* | *MCCS* |
| *d=10* | 0.3159 | **0.2791** | 0.7614 | 0.6663 |
| *d=30* | 0.3172 | **0.2706** | 0.7964 | 0.6438 |
| *d=50* | 0.3166 | **0.2636** | 0.8199 | 0.6305 |

nominal threshold, underscoring the inherent limitations of such heuristic strategies due to error propagation and affirming the theoretical soundness of our proposed approach.

# C    Details of Real Data Application

## C.1    Details of Tasks of *cv*

The *cv* tasks employ a pipeline for monocular depth estimation from RGB images, utilizing pre-trained ResNet and ViT models, respectively. This is combined with the *MCCS* approach based on conformal inference to identify samples falling within a specified target depth interval.

Experiments are conducted on the NYU Depth V2 dataset, which comprises 1,449 RGB-D images of indoor scenes along with depth maps captured using a Microsoft Kinect sensor. Each image has a resolution of 640×480 pixels, with depth values representing physical distances in meters. The dataset is partitioned into training (74%), calibration (13%), and test (13%) sets. Depth values are obtained by averaging the center crop of each depth map, resulting in a single scalar depth value per image. We then use a Ridge regressor to predict the depth for the following steps.

To intuitively demonstrate the applicability of our proposed method on *cv* tasks, Figure 8 illustrates the implementation workflow of the algorithm across different aspects of the dataset. The visualization component provides critical diagnostic and interpretive tools for evaluating the method's performance: 1) The *p*-value distribution histogram assesses conformity to uniform distribution assumptions under the null hypothesis; 2) The Benjamini-Hochberg procedure plot illustrates the relationship between sorted *p*-values and the critical threshold line, demonstrating the selection mechanism that controls false discoveries; 3) Depth distribution histograms contrast selected versus unselected samples, revealing how effectively the method identifies targets within the specified interval; 4) The prediction-versus-truth scatterplot visualizes regression accuracy while highlighting correct/incorrect selections relative to the interval boundaries. These visualizations collectively offer an assessment of both statistical properties (FDR control validity) and operational characteristics (depth estimation accuracy), enabling a comprehensive evaluation of the *MCCS*'s effectiveness in identifying samples within the target depth range. Figure 9 shows samples selected in the *cv* tasks.

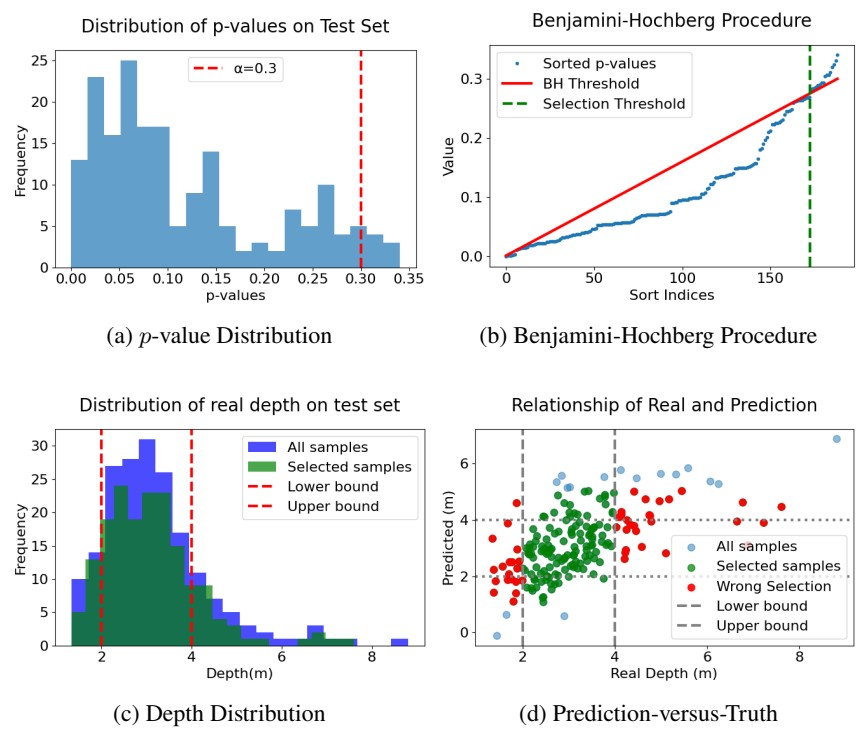

(a) *p*-value Distribution

(b) Benjamini-Hochberg Procedure

(c) Depth Distribution

(d) Prediction-versus-Truth

Figure 8: Comprehensive illustration of the implementation workflow on *cv* tasks.

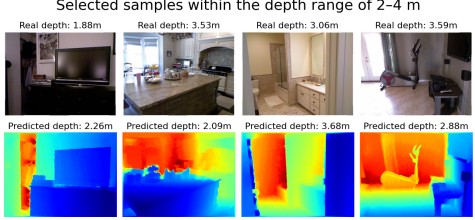

Figure 9: Selected Samples in the *cv* Tasks.

## C.2 DETAILS OF TASKS OF NLP

The *nlp* tasks employ a conformal prediction pipeline with FDR control to identify text prompts within a target toxicity range. Using text embeddings and regression models, toxicity scores are predicted; the *MCCS* method then selects samples within the desired interval (e.g., (0.12, 0.7)) while controlling errors. Applied to the Real-Toxicity-Prompts and Pile-Toxicity-Balanced datasets, the framework encompasses data loading, feature extraction via sentence embeddings, regression, nonconformity scoring, and the Benjamini–Hochberg procedure.

The Real-Toxicity-Prompts dataset contains 100K online texts, each annotated with continuous toxicity scores (0–1) from Perspective API, capturing diverse real-world language use. The Pile-Toxicity-Balanced subset includes 40K text segments from The Pile, with balanced binary toxicity labels (50% toxic / 50% non-toxic) achieved through stratified sampling to mitigate class imbalance. Together, these datasets support robust toxicity detection: the first enables fine-grained analysis of real-world toxicity, while the second ensures equitable evaluation via controlled class distribution.

The datasets are split via stratified sampling into training (60%), calibration (20%), and test (20%) sets to preserve toxicity distributions. For efficiency, up to 30,000 prompts are subsampled while maintaining score distribution. Text embeddings are generated using MiniLM-L6 (384-dimensional), a lightweight transformer, and used as features in a Ridge regression model trained only on the training set, with calibration and test sets reserved for subsequent steps. Figures 10 and 11 intuitively demonstrate our method's applicability by illustrating its workflow as Figure 8a and 8b.

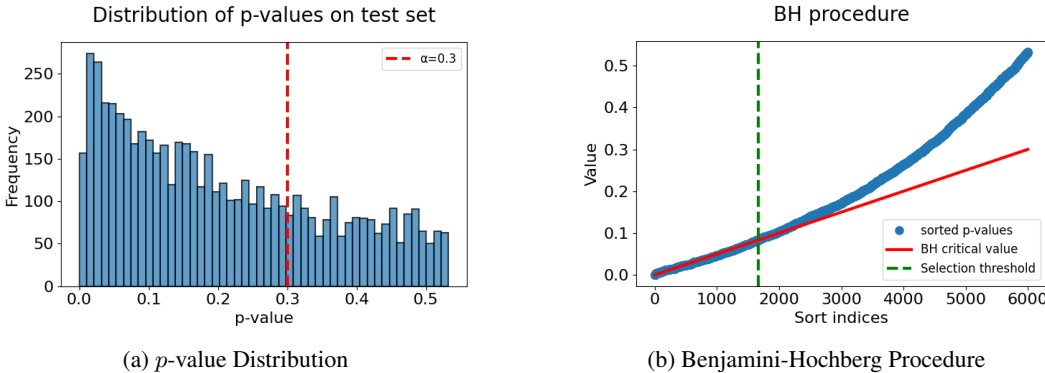

(a) *p*-value Distribution        (b) Benjamini-Hochberg Procedure

Figure 10: Illustration of the implementation workflow on *nlp* task of Real-Toxicity-Prompts.

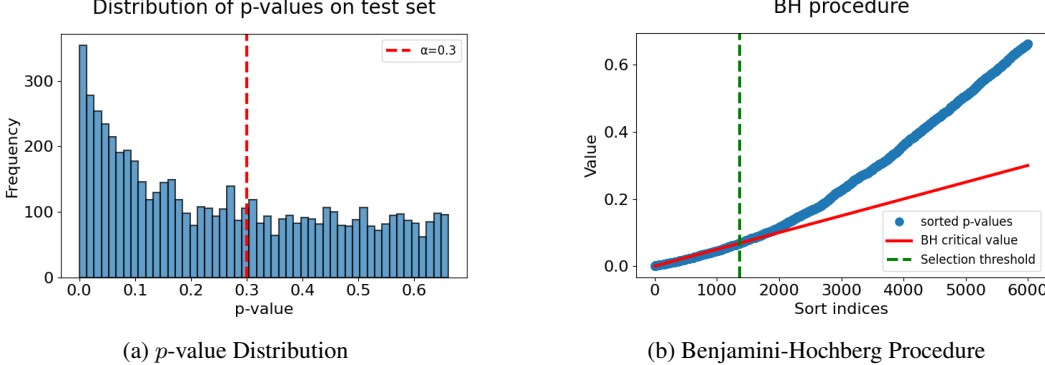

(a) *p*-value Distribution        (b) Benjamini-Hochberg Procedure

Figure 11: Illustration of the implementation workflow on *nlp* task of Pile-Toxicity-Balanced.

## C.3 DETAILS OF TASKS OF VQA

The VQA tasks are implemented through a pipeline that leverages conformal prediction to identify samples exhibiting high human annotation agreement within a predefined confidence interval. The

method specifically targets an agreement score range from 0.28 to 0.82, designed to capture instances with moderate to strong annotator consensus. At the core of the system, the BLIP vision-language model is employed to generate answer predictions along with confidence scores. These scores are subsequently mapped to estimates of human agreement using two distinct regression techniques, Ridge and BayesianRidge regression, each providing a complementary approach to modeling the relationship between model confidence and human consensus. The final selection of samples is performed using the *MCCS* (Multi-Class Calibrated Selection) method, which ensures rigorous control over the false discovery rate (FDR) throughout the inference process.

The VQA v2 dataset consists of 214,354 image-question pairs derived from the COCO 2014 validation set. Each question is associated with 10 human-annotated answers, facilitating the calculation of an agreement score, defined as the proportion of annotators who provided the most frequent response. This metric serves as a robust measure of consensus and label reliability. The implemented pipeline automatically downloads the COCO validation images, totaling 40,504, along with the corresponding questions and annotations. These components are processed through a customized VQA v2 Dataset class that systematically extracts and structures image-question-answer triplets, ensuring consistent and efficient data handling for subsequent modeling and analysis.

To ensure computational efficiency while preserving the essential statistical characteristics of the data, the implementation randomly samples a maximum of 1,000 image-question pairs. These samples are partitioned into training (80%), calibration (10%), and test (10%) subsets using random sampling, maintaining representative distributions across splits. The BLIP-VQA base model generates answer predictions along with confidence scores, obtained by averaging token probabilities during the text generation process. Meanwhile, human agreement scores are quantitatively defined as $\text{agreement} = \frac{\text{count of the most frequent answer}}{10}$, reflecting the consensus of annotators for each question. The implementation workflow for the *vqa* tasks is visually summarized in Figure 12, which follows the same structured format as Figure 8 to facilitate comparison and interpretation, thereby illustrating the method's practical applicability and scalability. Additionally, Figure 13 presents concrete examples of selected samples from the VQA tasks.

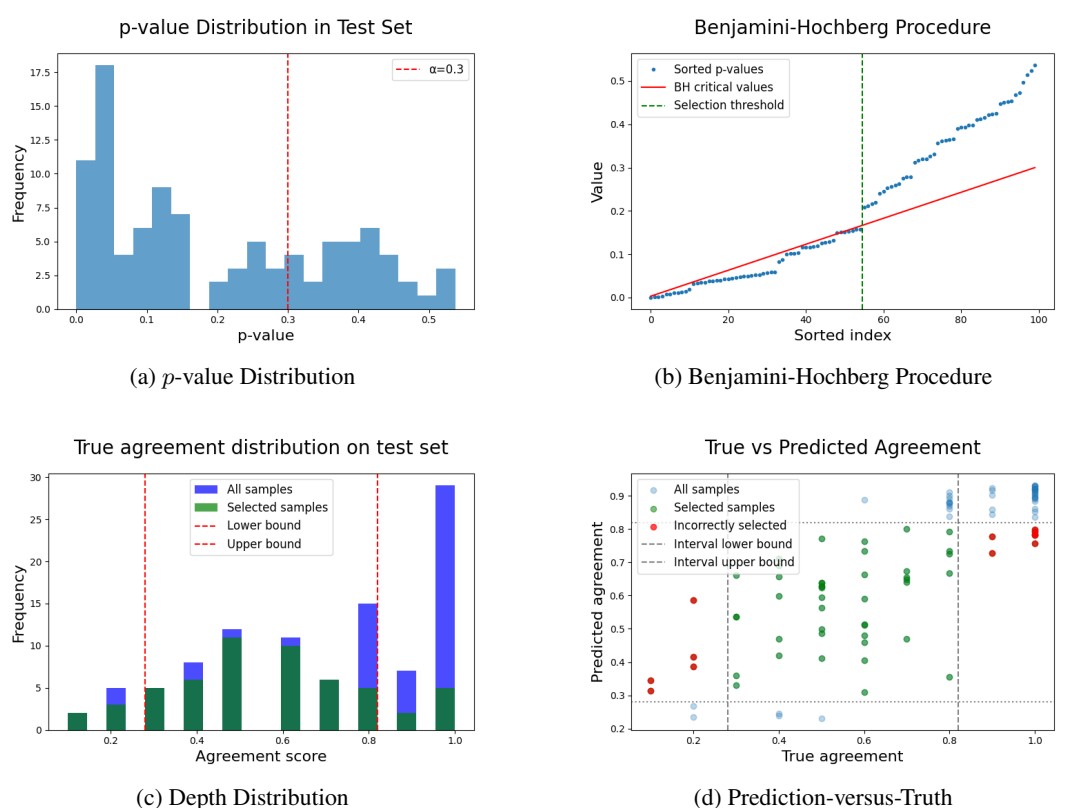

(a) *p*-value Distribution

(b) Benjamini-Hochberg Procedure

(c) Depth Distribution

(d) Prediction-versus-Truth

Figure 12: Comprehensive illustration of the implementation workflow on *vqa* tasks.

**Selected samples with moderately ambiguity**

Figure 13: Selected Samples in the *vqa* Tasks.

# D DETAILS OF MULTI-CLASS TASKS

## D.1 SELECTION FOR SINGLE-CLASS

The implementation of the *class* tasks on the CIFAR-10 dataset utilizes either ResNet or ViT architectures with FDR control in the *MCCS* approach to select specific images from multiple classes. The method accurately identifies target category samples (e.g., airplanes) while guaranteeing that the FDR remains below a predefined threshold, making it suitable for scenarios where only partial feature information of test samples is available. The core idea involves treating the multidimensional response not as a set of individually meaningful dimensions but rather as a holistic feature vector, thereby overcoming limitations inherent in existing approaches.

The CIFAR-10 dataset comprises 60,000 $32 \times 32$ RGB images across 10 classes: airplanes, automobiles, birds, cats, deer, dogs, frogs, horses, ships, and trucks, with 6,000 images per class, divided into 50,000 training and 10,000 test images. Data are partitioned via stratified sampling to preserve class balance: the test set contains 500 samples with equal representation of target/non-target classes, while the calibration set (also 500 samples) maintains $1 : 3$ target-to-non-target ratio of CIFAR-10. The remaining samples comprise the training set, ensuring an adequate number of positive examples for prototype computation while maintaining realistic class distributions.

The class prototyping approach establishes discriminative decision boundaries within the fine-tuned feature space, drawing inspiration from ProtoNet by representing each class through an embedded prototype. The target class prototype $\vec{p}$ is computed as the centroid of all target-class samples: $\vec{p} = \frac{1}{n} \sum_{i=1}^{n} \vec{f_i}$. A class-specific radius $r$, predefined according to the feature dimension, defines the acceptance region, with sample membership determined by the Euclidean distance metric $\left\| \vec{f} - \vec{p} \right\|_2$.

The regression component bridges the representational gap between feature domains via a Multi-OutputRegressor with Ridge regression. This model learns a mapping: $\mathbb{R}^{d_{\text{pretrained}}} \rightarrow \mathbb{R}^{d_{\text{fine-tuned}}}$ from pre-trained to fine-tuned feature spaces. Crucially, this mapping simulates the transformation from accessible features (e.g., pre-trained embeddings) to ideal but often unattainable features (e.g., fully fine-tuned representations), which can be substituted as needed in practical applications. Trained exclusively on the training partition, the regression model enables the inference-time prediction of task-adapted features from generic representations, enhancing applicability in real-world scenarios. By ensuring that the proportion of incorrectly selected samples remains below a predefined threshold (e.g., q = 0.1), this pipeline enables the reliable deployment of high-confidence predictions.

## D.2 Selection for Multi-Class

The selection of multiple target classes, representing disjunctive conditions, naturally extends the application of *MCCS* in multi-class tasks. This approach facilitates the extension from single-class to multi-class selection via parallel hypothesis testing using class-specific prototypes. The proportion of target class samples in the calibration set is set to 0.65 to ensure adequate positive samples for prototype computation. Figure 14 illustrates sample selection outcomes using the target class of "airplanes" in the Single-Class Task and the target classes of "airplanes, automobiles, and birds" in the Multi-Class Task as examples.

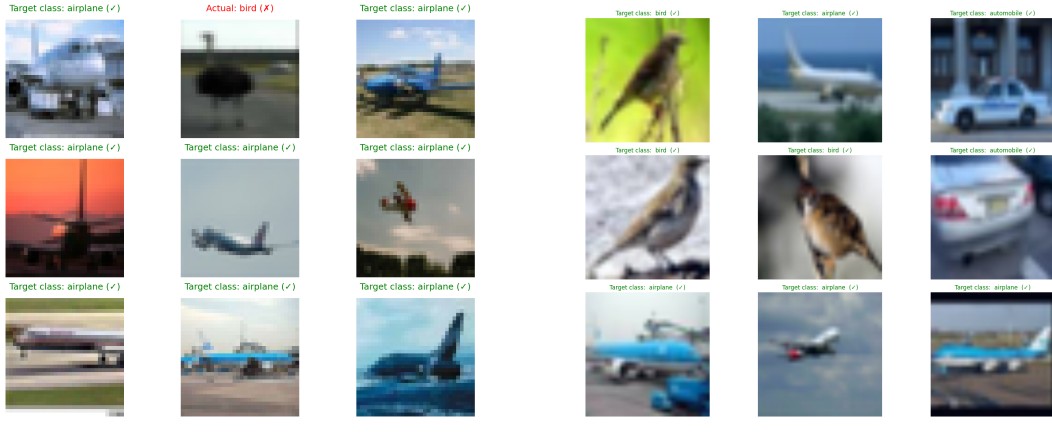

(a) Class "airplanes".                    (b) Classes "airplanes, automobiles, and birds".

Figure 14: Selected Samples in the Single-Class and Multi-Class Task.

## D.3 Selection for Similar-Class

The selection of similar target classes, representing conjunctive conditions, naturally extends the application of *MCCS* in multi-class tasks as identifying samples within a specific distance threshold from a given class prototype. This experiment utilizes the CIFAR-100 dataset due to its inherent superclass structure, which provides ground-truth semantic groupings for evaluation. The proportion of target class samples in the calibration set was set to 0.5 to ensure balanced representation.

To summarize, the *MCCS* on multi-class tasks is validated using the CIFAR-10 (Krizhevsky et al., 2009a) and CIFAR-100 (Krizhevsky et al., 2009b) datasets, enabling the selection of individual (*class-A*), multiple (*class-B*), and similar (*class-C*) classes. Results at a nominal FDR level of 0.1 are presented in Table 13. Our method maintains rigorous FDR control across all the *class* tasks.

Figure 15 displays a selection of retrieved samples for the query class "dolphin" within the target superclass "aquatic mammals," and the class proportion within the selected samples. Notably, some incorrectly selected samples, such as "shark", which does not belong to the superclass yet exhibits visual similarity to dolphins, were also identified, illustrating the method's ability to capture perceptually coherent features despite occasional semantic mismatches, thereby underscoring the rationale behind the approach. Furthermore, the balanced selection across categories rather than focusing on a single class demonstrates the method's effectiveness in handling similarity, a property ensured by the circular region defined around class prototypes.

Table 13: Observed FDR and power for *class* Tasks at the nominal level 0.1.

| Task | FDR | Power |
|---|---|---|
| *class-A* | 0.084 | 0.994 |
| *class-B* | 0.094 | 0.733 |
| *class-C* | 0.096 | 0.643 |

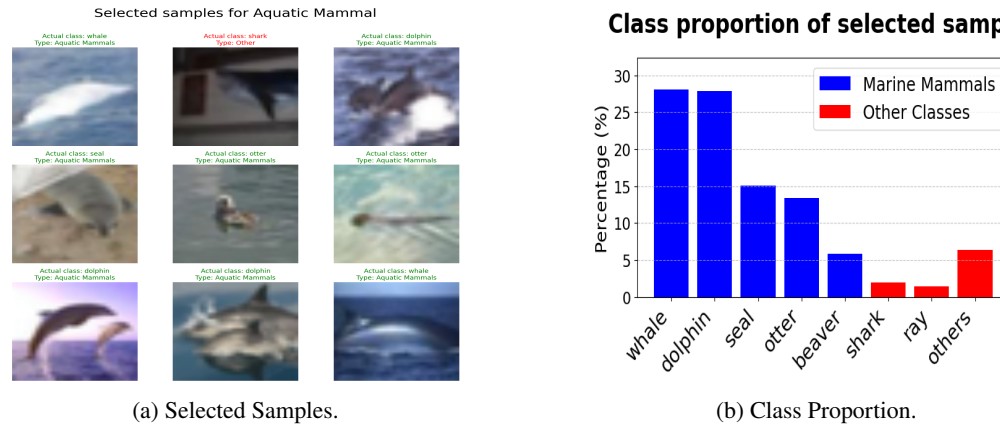

(a) Selected Samples.          (b) Class Proportion.

Figure 15: Selected Samples and Class Distribution in the Similar-Class Task.

# E ACKNOWLEDGMENT OF THE USE OF LARGE LANGUAGE MODELS (LLMS)

Large Language Models (LLMs) were used to aid in the writing and polishing of the manuscript. Specifically, we used an LLM to assist in refining the language, improving readability, and ensuring clarity in various sections of the paper. The model helped with tasks such as sentence rephrasing, grammar checking, and enhancing the overall flow of the text.

It is important to note that the LLM was not involved in the ideation, research methodology, or experimental design. All research concepts, ideas, and analyses were developed and conducted by the authors. The contributions of the LLM were solely focused on improving the linguistic quality of the paper, with no involvement in the scientific content or data analysis.

The authors take full responsibility for the content of the manuscript, including any text generated or polished by the LLM. We have ensured that the LLM-generated text adheres to ethical guidelines and does not contribute to plagiarism or scientific misconduct.

