# OpenReview forum: "Multi-Condition Conformal Selection"
_ICLR.cc/2026/Conference — ICLR 2026 Poster_

### Official Review · Reviewer_2sng · 2025-10-20

**Soundness:** 3
**Presentation:** 3
**Contribution:** 3
**Rating:** 8
**Confidence:** 3

**Summary:**

The paper introduces multi-conditional score for conformal selection (MCCS). Current approaches are restricted to single-threshold scenarios and MCCS proposes a novel nonconformity score for conjunctive and disjunctive conditions that is capable of dealing with multiple conditions. The experiments show the effectiveness of the technique in multiple applications and also for multi-variate scenarios.

**Strengths:**

- The paper presents a novel extension of conformal selection to multiple conditions, it addresses an important gap in the area.
- The paper present theoretical guarantees via finite sample FDR control.
- MCCS general framework is capable of handling both conjunctive and disjunctive conditions, multiple intervals and multivariate responses. That broad scope is a strength of the work.

**Weaknesses:**

- The experimentation section in terms of baselines look not fully explored. Experiments against other FDR control methods will strength the claims.

**Questions:**

- Can the authors add some discussion regarding computational requirements of MCCS, complexity and memory estimates will clarify and possibly amplify the applicability of the approach.
- Briefly discuss extensions of the current method to domain shift will strength the paper claims.
- Regarding algorithm 2 (Global BH procedure) $m \times K$ sorting seems a bottleneck, is there a way to make that part more efficient?

---

> ### Author Response · Authors · 2025-11-25
> **Response to Reviewer 2sng (1/2)**
>
> Thanks for your recognition and the valuable suggestions. Please find our response below.
>
> ### **1. Experiments against other FDR control methods. [W1]**
>
> Thank you for raising this concern. As suggested, we evaluated two additional baseline methods, Benjamini-Yekutieli (BY) [1] and Prediction Interval (PI) [2], to strengthen the comparison with MCCS. While existing approaches are designed for single-condition settings, we adapted $BY$ and $PI$ for multi-condition selection by incorporating appropriate modifications. The $BY$ procedure addresses potential $p$-value dependencies via a harmonic correction, adjusting the rejection threshold to $\frac{q \cdot l}{m \cdot K \cdot c(m)}$, where $c(m) = \sum_{i=1}^m 1/i$. The $PI$ method relies on parametric prediction intervals (e.g., under normal errors) to select samples whose intervals overlap with the target region, such as by checking $[L_j, U_j] \cap I_{\text{Target}} \neq \emptyset$ for each test sample. We compared these with MCCS on Tasks $1$-$6$ under Setting $1$ and noise level $0.5$, with results in the table below(best values in bold).
>
>
> | Task | |  | FDR| | |  |Power | |
> |:-:|:-: |:-:|:-:|:-:|:-:|:-:|:-:|:-:|
> | |  |*BY* | *PI* | *MCCS*  || *BY* | *PI* | *MCCS* |
> | *1* | | 0.0353 | 0.5039 | **0.2987**  | | 0.6242 | 0.8906 | **0.9954** |
> | *2* | | 0.0369 | 0.4057 | **0.2995**  | | 0.6522 | 0.9848 | **0.9952** |
> | *3* | | 0.0335 | 0.4436 | **0.3037**  | | 0.2181 | 0.9652 | **0.9697** |
> | *4* | | 0.0375 | 0.4106 | **0.3009**  | | 0.2452 | **0.9942** | 0.9732 |
> | *5* | | 0.0243 | 0.3562 | **0.2965**  | | 0.1428 | **0.9641** | 0.9487 |
> | *6* | | 0.0239 | 0.4021 | **0.3030**  | | 0.2240 | 0.9692 | **0.9823** |
>
>
> As shown, $BY$ showed excessively low FDR and reduced power due to stringent thresholds. $PI$ achieved high power but consistently exceeded the FDR target, highlighting its robustness issues. Only MCCS maintained FDR near the nominal level with high power, confirming MCCS's superiority in multi-condition settings. We added this part to Appendix B.4 of the revised manuscript.
>
>
> [1] Benjamini, Y., & Yekutieli, D. (2001). THE CONTROL OF THE FALSE DISCOVERY RATE IN MULTIPLE TESTING UNDER DEPENDENCY. Annals of Statistics, 29, 1165-1188.
>
> [2] Lehmann, E. L., & Casella, G. (1998). Theory of point estimation(2nd ed.). Springer.
>
>
>
>
> ---
>
>
>
>
>
> ### **2. Computational requirements of MCCS. [Q1]**
>
> Thank you for highlighting this issue. MCCS extends the cfBH [1] framework to multi-condition settings, which introduces a linear increase in computational complexity relative to the number of intervals K. Specifically, the time complexity of MCCS is $O(m·n·K)$, driven by pairwise $p$-value computations for each of the $K$ intervals, compared to $O(m·n)$ for the base cfBH method (or $O((n+m)·log n$) with optimizations). Similarly, the space complexity is $O(m·K)$ for storing the $p$-value matrix, versus $O(n+m)$ for cfBH. This scaling is manageable in practice, as $K$ is often small compared to data sizes (e.g., in drug discovery or screening tasks), and the costs grow linearly without exponential bottlenecks. Empirical evaluations in Section 5.3, such as NLP and computer vision tasks, demonstrate its feasibility on real-world data.
>
> [1] Jin Y, et al. Selection by prediction with conformal p-values. JMLR2023.

---

> ### Author Response · Authors · 2025-11-25
> **Response to Reviewer 2sng (2/2)**
>
> ### **3. Extensions of the current method to domain shift. [Q2]**
>
>
> Thank you for the question. The issue of domain shift is complementary to the multi condition setting considered in our paper. In other words, our method can be integrated with existing conformal selection methods designed for domain shift (e.g, WCS [1]). For example, one could yield a weight function $w(x, y)$ to handle concept shift, while sliding windows or exponential weighting might address non-stationary distributions in dynamic environments. This extension can strengthen our method in those settings with domain shift, similar to vanilla conformal selection.
>
> [1] Jin, Y., & Candès, E.J. (2023). Model-free selective inference under covariate shift via weighted conformal p-values. Biometrika.
>
>
>
>
>
> ---
>
>
>
>
>
>
>
>
>
> ### **4. Optimization of Sorting Bottleneck [Q3]**
>
>
> Thank you for bringing this to our attention. The identified sorting bottleneck in the global BH procedure can be optimized using the Quickselect algorithm [1] . This approach, termed $MCCSqck$, reduces sorting overhead by directly determining the BH threshold, avoiding full sorting. We compared MCCS and MCCSqck across varying numbers of intervals under Setting $1$ and noise level $0.5$. Results are shown in the table below, where Time refers to the time taken for $100$ replications without GPU acceleration. The results demonstrate that MCCSqck maintains comparable FDR and Power metrics to MCCS, while runtime advantages become increasingly pronounced as $K$ grows, underscoring its reliability for large-scale applications. This optimization aligns with MCCS's rigorous framework, offering a practical balance between efficiency and statistical robustness. We added this part to Discussion of the revised manuscript.
>
>
>
> | $I_{Target}$ | |  | FDR| | |Power |  | |Time |
> |:-:|:-:|:-:|:-|:-:|:-:|:-|:-:|:-:|:-|
> | |  |*MCCS* | *MCCSqck* | |*MCCS*|*MCCSqck*||*MCCS*| *MCCSqck*|
> | *K=4* | | 0.3030 |0.2971 | | 0.9823 |0.9810 | | 54s | ${\bf 52s}^{\downarrow 2s}$|
> | *K=10* | | 0.2935 |0.2950 | | 0.9951|0.9951| |151s | ${\bf 124s} ^{\downarrow 27s}$ |
> | *K=20* | | 0.2941 |0.2945| |0.9901|0.9901| |294s| ${\bf 238s}^{\downarrow 56s}$|
> | *K=40* | | 0.2762 |0.2822| |0.9052|0.9610| |560s|${\bf 469s}^{\downarrow 91s}$|
>
>
>
>
> [1] Hoare, C. A. R. (1961). Algorithm 65: FIND. Communications of the ACM, 4(7), 321-322.

---

### Official Review · Reviewer_eRb5 · 2025-10-27

**Soundness:** 3
**Presentation:** 3
**Contribution:** 2
**Rating:** 4
**Confidence:** 4

**Summary:**

This paper introduces the Multi-Condition Conformal Selection (MCCS) algorithm, an extension of the conformal selection framework designed to address scenarios involving multiple selection criteria. The method specifically tackles conjunctive conditions and disjunctive conditions, which frequently arise in practical applications where decisions depend on simultaneous or alternative constraints. The central innovation lies in the development of a tailored nonconformity score that ensures regional monotonicity for interval-based selections, alongside a global adaptation of the Benjamini-Hochberg (BH) procedure to manage disjunctive constraints. These components are underpinned by rigorous theoretical analysis, establishing finite-sample control of the False Discovery Rate (FDR) under exchangeability assumptions.

**Strengths:**

1.  **Theoretical Soundness:** The paper provides rigorous theoretical foundations, including proofs for finite-sample FDR control under exchangeability conditions (Theorem 4.1). The extension of conformal selection theory to multi-condition settings is non-trivial and well-articulated.
2.  **Comprehensive Evaluation:** The experimental section is thorough, demonstrating the method's applicability across diverse modalities (text, images, multi-modal) and tasks (single-class, multi-class, similar-class selection). The comparison with relevant baselines (Inter-cfBH, Union-cfBH) is appropriate and highlights the limitations of naive approaches.
3.  **Clear Presentation:** The paper is generally well-written, with a logical flow from problem formulation to methodology, theory, and experiments. The figures, such as the illustrative diagrams, aid in understanding the core concepts.

**Weaknesses:**

1.  **Perceived Marginal Technical Contribution (Major Concern):** The fundamental idea can be viewed as generalizing the single-threshold selection $Y > c$ to a more general form $Y \in I$. A significant portion of the theoretical and methodological machinery seems to be a direct adaptation of the established cfBH (Jin & Candès, 2023) framework.
     - The problem could potentially be reframed by defining a suitable function $\phi(Y, I)$ that encapsulates the multi-condition logic, reducing it to a single-threshold problem on $\phi(Y, I)$. For instance, for an interval $I = (a, b)$, one could define $\phi(Y, I) = \min(b-Y, Y-a)$, where $\phi(Y, I) > 0$ is equivalent to $Y \in (a, b)$. A similar construction could be devised for disjunctive conditions. The core conformal p-value machinery and BH procedure from cfBH could then be applied to $\phi(Y, I)$, potentially obviating the need for the newly proposed nonconformity scores and the specific "global BH" narrative. While the authors' approach of designing a regionally monotone score is one valid path, the paper would be significantly strengthened by directly addressing this alternative, more unified perspective. A discussion justifying why their specific decomposition (separate scores per interval + global BH) is preferable to a unified $\phi$-function approach is crucial. The current presentation makes the contribution feel more like a specialized extension rather than a fundamental generalization.

2.  **Clarity on Theoretical Novelty:** The proof techniques, particularly for Theorem 4.1, heavily rely on and extend the principles established in cfBH. The paper could more clearly delineate the specific novel technical challenges overcome in the multi-condition setting compared to the single-condition base.

[1] Jin Y, Candès E J. Selection by prediction with conformal p-values[J]. Journal of Machine Learning Research, 2023, 24(244): 1-41.

**Questions:**

1. **On the Core Technical Contribution and Methodological Necessity:** My primary concern pertains to the fundamental architecture of MCCS. Please refer to weakness 1.
2. **On Experimental Design and Interpretation:**
    - The experiments primarily feature a small number of conditions (e.g., Tasks 1-6 in Table 2). A key claim of the method is its generalizability to diverse condition combinations. How does the power of MCCS scale as the number of intervals $K$ becomes large?
    - The paper reports aggregate FDR and power. It would be insightful to analyze which specific conditions (which intervals ) contribute most to the discoveries. Does the global BH procedure lead to a balanced selection across all target intervals, or does it favor intervals with certain characteristics?
3. **On the Equivalent Interval Representations:** Consider a target interval $I = (a, b)$. This can be equivalently expressed as a single interval or decomposed into two sub-intervals $(a, c] \cup [c, b)$ (for $K=2$). While these two representations are semantically equivalent in defining the target set, will they lead to different algorithmic constructions? Can you explain on this?

I will raise my score if these concerns are properly handled.

---

> ### Comment · Reviewer_eRb5 · 2025-11-25
>
> As the review deadline is approaching, we kindly request a prompt reply to allow sufficient time for any further discussions or clarifications that may be needed.

---

> > ### Author Response · Authors · 2025-11-25
> > **Response to Reviewer eRb5 （1/2）**
> >
> > We sincerely apologize for the delayed response and greatly appreciate your recognition of our work and your valuable suggestions. Please find our response below.
> >
> > ### **1. On the necessity of the method design. [W1, Q1]**
> >
> > Thank you for this insightful concern. A method similar to $\phi(Y, I)$ has already been used as a baseline, which has been analyzed in Appendix B.2. We further provide a detailed explanation in the General Response.
> >
> >
> >
> > ---
> >
> >
> >
> >
> > ### **2. Clarity on Theoretical Novelty.[W2]**
> >
> > Thank you for allowing us to clarify the novelty of our method. While MCCS builds on cfBH’s proof framework, its core theoretical contribution lies in extending this foundation to address novel challenges unique to multi-condition settings, challenges absent in single-condition frameworks like cfBH.
> >
> > * First, MCCS generalizes the hypothesis space from binary per-sample decisions to a composite structure with K distinct null hypotheses per sample (e.g., $H_0^{j,k}: y_{n+j} \notin I_k$). This requires managing a global set of $m \times K$ hypotheses, enabling structured multi-way selection.
> > * Second, the proof handles complex dependencies: $p$-values for the same sample $j$ are block-dependent due to shared covariates, and MCCS rigorously establishes that the BH procedure remains valid under this structure.
> > * Third, the design of a non-conformity score with regional monotonicity (e.g., $V^k(x, y)$ in Algorithm 1) is critical; it ensures stochastic dominance of oracle $p$-values, a property not directly pointed out in cfBH.
> > * Finally, MCCS proves that the global BH procedure controls FDR for the entire hypothesis set, despite dependencies and composite intervals.
> >
> > In summary, MCCS’s novelty stems from solving these multi-condition-specific challenges within a conformal selection framework. These extensions are non-trivial and form the basis of MCCS’s novelty.
> >
> >
> >
> >
> >
> >
> >
> > ---
> >
> >
> >
> >
> >
> >
> >
> > ### **3. Performance under larger K. [Q2.1]**
> >
> >
> >
> >
> > Thank you for the suggestion to strengthen the experiments. Our tasks designed in Section 5 were primarily intended to demonstrate various combinatorial scenarios without accounting for the number of subintervals. As you suggested, we extended the evaluation by conducting additional experiments with a significantly larger value of K (an order of magnitude higher than in Section 5). In these experiments, the target interval was defined as $I_{\text{Target}} = \bigcup_{k=1}^{K} (c_{k_L}, c_{k_R})$, with the setting fixed at $1$ and noise levels varied at $0.1$, $0.5$, and $0.9$, respectively. As shown in the table below, as the number of intervals $K$ increases, statistical power decreases slightly while remaining high, and FDR control becomes more conservative while staying near the nominal level. This occurs because the BH procedure's critical value $\frac{q \cdot l}{m \cdot K}$ causes rejection regions to contract, reducing true signal detection, while the increased proportion of null hypotheses due to interval subdivision necessitates more conservative thresholds to maintain overall FDR control. These performance changes under large $K$ represent natural consequences of multiple testing adjustments, demonstrating MCCS's statistical rigor. We added this part to Section 5.2 of the revised manuscript.
> >
> > | Noise Level | |  | FDR| | |  |Power | |
> > |:-:|:-:|:-:|:-:|:-:|-|:-:|:-:|:-:|
> > | |  |*K=10* | *K=20* | *K=40*  || *K=10* | *K=20* | *K=40* |
> > | *Ns=0.1* | | 0.2926 | 0.2886 | 0.2773  | | 0.9942 | 0.9789 | 0.9069 |
> > | *Ns=0.5* | | 0.2935 | 0.2882 | 0.2762  | | 0.9951 | 0.9785 | 0.9052 |
> > | *Ns=0.9* | | 0.2951 | 0.2888 | 0.2759  | | 0.9946 | 0.9777 | 0.9038 |

---

> > > ### Author Response · Authors · 2025-11-25
> > > **Response to Reviewer eRb5 （2/2）**
> > >
> > > ### **4. Analysis of interval-specific contributions. [Q2.2]**
> > >
> > >
> > > Thank you for the suggestion to extend our analysis. In MCCS, samples are uniformly sorted by their conformal $p$-values during the global BH procedure. If selections favor certain intervals, this reflects stronger statistical evidence, such as greater conformity to target regions, due to inherent variations in the data distribution. To illustrate this point, we examined the selection proportions across intervals in Task $6$, using Settings $1$ and $3$ as representative examples. The table below compares the proportions of all test samples and of selected test samples in each target interval. The results reveal a consistent distribution pattern of $All$ and $Selected$ samples across intervals. (Note that the sum of selection proportions exceeds $1−q$ because Task $6$ involves overlapping intervals.) We added this part to Section 5.2 of the revised manuscript.
> > >
> > > | *k* | | | *ST1* |  | || *ST3* |  |
> > > |:-:|-|-|:-|:-|-|-|:-|:-|
> > > |  | | | *All* | *Selected* | ||*All* | *Selected* |
> > > | *1* | | | ░░ 12.5%| ▓▓▓▓ 24.6% | | |░ 5.9%| ▓▓ 13.1% |
> > > | *2* | | |  ░░░░ 22.2%| ▓▓▓▓▓▓ 36.8% | | |░ 3.6%| ▓ 9.6% |
> > > | *3* | | | ░░ 10.8%| ▓▓ 14.1% | | |░ 6.4%| ▓ 9.4% |
> > > | *4* | | |  ░░ 12.3%| ▓▓▓▓ 24.5% | | |░░░░░░░░ 42.8%| ▓▓▓▓▓▓▓▓▓▓ 68.0% |
> > >
> > >
> > >
> > >
> > >
> > >
> > >
> > >
> > >
> > > ---
> > >
> > >
> > >
> > >
> > >
> > >
> > >
> > >
> > > ### **5. On the Equivalent Interval Representations. [Q3]**
> > >
> > >
> > > Thank you for this thoughtful observation. In MCCS, equivalent interval representations, such as a single interval versus decomposed sub-intervals, yield different selection sets due to differences in hypothesis-testing granularity, but both ensure FDR control. The single-interval approach uses one $p$-value per sample, while sub-intervals generate multiple $p$-values, altering the BH procedure thresholds. Nonetheless, FDR is maintained rigorously. MCCS supports both representations, allowing users to choose based on coarse or fine-grained selection needs, highlighting its flexibility for multi-condition scenarios.

---

> > > ### Comment · Reviewer_eRb5 · 2025-11-26
> > >
> > > Regarding the IND results, several points require clarification:
> > >
> > > 1. The construction of $f$ in Appendix B.2 appears unusual. When multiple intervals are widely separated, their product may unduly influence behavior near individual interval boundaries. Consider defining $f(Y, I)$ as the signed distance from $Y$ to the nearest endpoint of interval $I$ (positive inside intervals, negative outside). This approach would prevent distant intervals from affecting local behavior and intuitively should perform similarly to the proposed method. Please explain why the reported power is 0.0000 and specify the sample size used.
> > >
> > > 2. Line 376 claims that "the IND approach performs strongly, depending on the accuracy of the fitted model." For selection tasks, power would naturally be low if prediction accuracy is poor. How does the proposed method achieve reduced dependency on prediction quality compared to this approach?

---

> > > > ### Comment · Reviewer_eRb5 · 2025-11-26
> > > >
> > > > Typically, we do not set the FDR level at 0.3. Have you experimented with its performance at a level of 0.1?

---

> ### Author Response · Authors · 2025-11-27
> **Response to Reviewer eRb5 (1/2)**
>
> Thank you for raising these concerns, which provide us with a valuable opportunity to further clarify our work. Our responses are summarized below:
>
> ### **1. The variant with signed distance on disjunctive conditions**
>
> Thank you for the suggestion. We conduct additional experiments with the method of modifying $f$ with "signed distance" to mitigate the impact of the "distant intervals", denoted as $Ind_{eRb5}$. The two tables below present the empirical results of three methods under disjunctive (i.e., $(-\infty，-20) \cup (60，+\infty)$) and moderate-disjunctive settings (i.e., $(-\infty，-4) \cup (60，+\infty)$), respectively. The results show that our method significantly outperforms both $Ind$ and $Ind_{eRb5}$ with preciser FDR control and higher power, while $Ind_{eRb5}$ indeed achieves better performance than $Ind$.
>
>
>
> **Why the reported power is 0?** The results are reasonable as the performance of $Ind$ and $Ind_{eRb5}$ are very poor in the disjuctive setting with widely separated intervals (i.e., $(-\infty，-20) \cup (60，+\infty)$). We found that these two methods provide either empty set (FDR=0) or a few cadidates that do not statify the condition (FDR=100%). Similar results are also reported in Table 5 of mCS [1], where the $bi$ method (similar to $Ind$) achieves powers of $0$ in all tasks. Besides, we conducted additional experiments under a moderate-disjunctive setting (i.e., $(-\infty，-4) \cup (60，+\infty)$) . The results show that $Ind$ and $Ind_{eRb5}$ can achieve non-zero powers in the moderate-disjunctive setting but are still suboptimal compared to our method.
>
>
> **The sample size used.** In this work, we use a sample size of $200$ for the test sizes. Following cfBH [2] and mCS [1], we provid extensive results with sample sizes of {200, 500, 1000} in the new results provided in this response. The results show that our method consistently achieves the best performance across various settings of sample size.
>
>
> **Results at various FDR levels.** In the manuscript, we reported results for $q=0.3$ following the setting in mCS [1], and presented results of MCCS at various $q$-levels (ranging from 0.05 to 0.5 inincrements of 0.05) in **Section 5.2**. Following the reviewer's suggestion, we report extensive results at various FDR levels (including 0.1, 0.3, 0.5) in all tables of this response, which validate the consistent advantage of our method.
>
> We add these new results in Appendix B.3 of the revised version.
>
> **Results on disjunctive conditions**
> | **TestSize** | **$q$** |  |  | **FDR** |  | | | **Power** |  |
> |:--------:|:--------:|:--------:|:--------:|:--------:|:--------:|:--------:|:--------:|:--------:|:---------:|
> |  |  |  | *Ind* | $Ind_{eRb5}$ | *MCCS* | | *Ind* | $Ind_{eRb5}$ | *MCCS*|
> | *200* | *0.1* |  | 0.0600 | 0.1200 | 0.0661 | | 0.0000 | 0.0000 | 0.6027 |
> | *200* | *0.3* |  | 0.2290 | 0.2400 | 0.2848 | | 0.0000 | 0.0000 | 0.9515 |
> | *200* | *0.5* |  | 0.2700 | 0.3300 | 0.4893 | | 0.0000 | 0.0000 | 0.9965 |
> | *500* | *0.1* |  | 0.0500 | 0.1000 | 0.0968 | | 0.0000 | 0.0000 | 0.8213 |
> | *500* | *0.3* |  | 0.1700 | 0.2100 | 0.2972 | | 0.0000 | 0.0000 | 0.9775 |
> | *500* | *0.5* |  | 0.3000 | 0.3600 | 0.4958 | | 0.0000 | 0.0000 | 0.9981 |
> | *1000* | *0.1* |  | 0.0900 | 0.0500 | 0.0967 | | 0.0000 | 0.0000 | 0.8782 |
> | *1000* | *0.3* |  | 0.2400 | 0.2100 | 0.2999 | | 0.0000 | 0.0000 | 0.9833 |
> | *1000* | *0.5* |  | 0.3000 | 0.3200 | 0.4928 | | 0.0000 | 0.0000 | 0.9993 |
>
>
> **Reults on moderate-disjunctive conditions**
> | **TestSize** | **$q$** |  |  | **FDR** |  | | | **Power** |  |
> |:--------:|:--------:|:--------:|:--------:|:--------:|:--------:|:--------:|:--------:|:--------:|:---------:|
> |  |  |  | *Ind* | $Ind_{eRb5}$ | *MCCS* | | *Ind* | $Ind_{eRb5}$ | *MCCS*|
> | *200* | *0.1* |  | 0.0663 | 0.0793 | 0.0929 | | 0.1869 | 0.1770 | 0.9787 |
> | *200* | *0.3* |  | 0.2583 | 0.1888 | 0.2886 | | 0.6039 | 0.5081 | 0.9998 |
> | *200* | *0.5* |  | 0.3931 | 0.4130 | 0.4869 | | 0.7105 | 0.7734 | 1.0000 |
> | *500* | *0.1* |  | 0.0619 | 0.0634 | 0.1001 | | 0.2923 | 0.3158 | 0.9955 |
> | *500* | *0.3* |  | 0.2204 | 0.2359 | 0.2959 | | 0.6388 | 0.6229 | 1.0000 |
> | *500* | *0.5* |  | 0.4592 | 0.3892 | 0.4922 | | 0.8670 | 0.7378 | 1.0000 |
> | *1000* | *0.1* |  | 0.0928 | 0.0559 | 0.1014 | | 0.3631 | 0.3390 | 0.9972 |
> | *1000* | *0.3* |  | 0.2624 | 0.2889 | 0.2997 | | 0.7100 | 0.7448 | 0.9999 |
> | *1000* | *0.5* |  | 0.4613 | 0.4741 | 0.4982 | | 0.8785 | 0.9039 | 1.0000 |

---

> > ### Author Response · Authors · 2025-11-27
> > **Response to Reviewer eRb5 (2/2)**
> >
> > **Reults on conjunctive conditions**
> >
> > | **TestSize** | **$q$** |  |  | **FDR** |  | | **Power** |
> > |:--------:|:--------:|:--------:|:--------:|:--------:|:--------:|:--------:|:--------:|
> > |       |       |  | *Ind* | *MCCS* | | *Ind* | *MCCS* |
> > | *200* | *0.1* |  | 0.0700 | 0.0952 |  | 0.0358 | 0.7158 |
> > | *200* | *0.3* |  | 0.2013 | 0.2126 | | 0.2874 | 0.9756|
> > | *200* | *0.5* |  | 0.3223 | 0.4895 | | 0.3222 | 1.0000 |
> > | *500* | *0.1* |  | 0.1100 | 0.0960 | | 0.0723 | 0.9460 |
> > | *500* | *0.3* |  | 0.1474 | 0.2987 | | 0.2465 | 0.9978 |
> > | *500* | *0.5* | | 0.4055 | 0.4995 | | 0.5006 | 1.0000 |
> > | *1000* | *0.1* | | 0.0693 | 0.0967 | | 0.1353 | 0.8786 |
> > | *1000* | *0.3* |  | 0.2209 | 0.2993 | | 0.3508| 0.9980 |
> > | *1000* | *0.5* |  | 0.3297 | 0.4997 | | 0.5110 |1.0000 |
> >
> >
> > ---
> >
> > ### **2. The dependence on the predictor performance**
> >
> >
> > Thank you for pointing our the ambiguous claim. We clarify that this claim is a **conjecture** to explain the empirical results: the extremely low power of the $Ind$ approach can be attributed to its poor performance in predicting the transformed target. In other words, we guess that the predictive performance of models trained on the transformed target will be inferior to that of models trained on the original target, leading to lower power of the $Ind$ approach. This is supported by the empirical results shown in the table below: the $Ind$ approach achieves lower $R^2$ than our method, as well as power. This aligns with the established principle that "power would naturally be low if prediction accuracy is poor for selection tasks". To mitigate the ambiguity, we **remove this conjecture in the revised manuscript** since it cannot explain the performance under the moderate-disjunctive setting. In addition to this, we conjecture that the superior performance of our method might come from its aggregating evidence from multiple $p$-values of various conditions, instead of relying on a single result of a transformation.
> >
> >
> >
> >
> > | | |disjunctive||| | moderate| |
> > |:-:|:-:|:-:|:-:|:-:|:-:|:-:|:-:|
> > | | $R^2$ |FDR| Power ||$R^2$ |FDR| Power |
> > | Ind | 0.3230 |0.0600 |0.0000||0.1986|0.0743 |0.3422|
> > | MCCS | 0.8758 | 0.0661 | 0.6027 ||0.8758|0.0929|0.9787 |
> >
> >
> >
> >
> > [1] Tian B, et al. Multivariate Conformal Selection. ICML 2025.
> >
> > [2] Jin Y, et al. Selection by prediction with conformal p-values. JMLR 2023.

---

> > > ### Comment · Reviewer_eRb5 · 2025-11-28
> > >
> > > Your experimental results are impressive and partially convince me of your method's effectiveness. However, I still struggle to understand why the IND approach performs so poorly. Additionally, as reviewer GiWV pointed out, ACS is fully applicable to this problem (though they haven't run experiments), and the requirement for $Y$ to be scalar is unnecessary, the theoretical framework holds without this assumption. Beyond experimental comparisons, I would like a theoretical explanation of your method's advantages. Relying solely on experimental results is not fully persuasive when existing solutions already address the problem.

---

> > > > ### Author Response · Authors · 2025-12-03
> > > > **Response to Reviewer eRb5 (1/2)**
> > > >
> > > > ### **1. Superiority of MCCS over $\textit{Ind}$**
> > > >
> > > >
> > > >
> > > > Thank you for the opportunity to explain the superiority of our method further. As we explained in our previous response, our method considers the $p$-value of each interval, thereby not only improving our overall performance but also allowing us to control the selection ratio for each interval, which $Ind$-like methods, which treat all conditions uniformly, cannot do. As you suggested, we hereby theoretically demonstrate that this fine-grained control will maintain FDR control.
> > > >
> > > >
> > > >
> > > >
> > > > **1. Problem setup**
> > > >
> > > > In MCCS, we consider $m$ test samples and $K$ target intervals, resulting in a total of $N = m \times K$ hypothesis tests. Each hypothesis $H_{jk}$ corresponds to test sample $j$ and target interval $I_k$.
> > > >
> > > > We introduce predefined and data-independent weights $w_{jk} > 0$ to reflect the importance or prior information of different hypotheses, and define the weighted $p$-value as:
> > > >
> > > > $p_{jk}^{\text{weighted}} = \frac{p_{jk}}{w_{jk}}$
> > > >
> > > > **2. Conditions Required for Weights**
> > > >
> > > > To ensure FDR control, the weights must satisfy the following conditions:
> > > >
> > > > 1. Non-negativity: $w_{jk} > 0$ for all $j$, $k$.
> > > > 2. Normalization: $\frac{1}{N}\sum_{j=1}^m \sum_{k=1}^K w_{jk} = 1$, i.e., $\sum_{j,k} w_{jk} = N$.
> > > > 3. Null Hypothesis Weight Constraint: $\sum_{(j,k) \in \mathcal{H}\_0} w_{jk} \leq N$, where $\mathcal{H}_0$ is the set of null hypotheses (hypotheses where the true condition does not hold).
> > > >
> > > > Note: Condition *3* is equivalent to requiring that the average weight of null hypotheses does not exceed *1*, but this is automatically satisfied because:
> > > > * From Condition *2*, the average weight of all hypotheses is $1$.
> > > > * The sum of weights for null hypotheses $\le$ the total weight sum = $N$.
> > > > * Thus, $\frac{\sum_{\mathcal{H}\_{0}} w_{jk}}{\mathcal{H}\_{0} } \leq \frac{N}{ \mathcal{H}\_{0}
> > > > } \leq \frac{N}{1} = N$ (though this is not strictly needed).
> > > >
> > > > In practice, Condition *3* is automatically satisfied because the sum of weights for null hypotheses cannot exceed the total weight sum $N$.
> > > >
> > > > **3. Weighted BH Algorithm**
> > > >
> > > > The algorithm steps are as follows:
> > > >
> > > > 1. Compute the weighted $p$-values: $p_{jk}^{\text{weighted}} = p_{jk} / w_{jk}$.
> > > > 2. Sort the weighted $p$-values in ascending order: $p_{(1)}^{\text{weighted}} \leq p_{(2)}^{\text{weighted}} \leq \cdots \leq p_{(N)}^{\text{weighted}}$.
> > > > 3. Find the largest index $l$ such that: $p_{(l)}^{\text{weighted}} \leq \frac{q \cdot l}{N}$.
> > > > 4. Reject the corresponding $l$ hypotheses.
> > > >
> > > >
> > > > **4. Proof Process**
> > > >
> > > > 4.1. **Notation Definitions**
> > > >
> > > > * $\mathcal{H}\_0$: Set of null hypotheses $(y_{n+j} \notin I_k)$.
> > > > * $R_{jk} = \mathbb{I}(H_{jk} \text{ is rejected})$.
> > > > * $\mathcal{S} = \{(j,k): R_{jk} = 1\}$, $R=\mathcal{S}.$
> > > > * $V = \mathcal{S} \cap \mathcal{H}_0$ (number of false discoveries).
> > > > * $\text{FDR} = \mathbb{E}\left[\frac{V}{R \vee 1}\right]$.

---

> > > > > ### Author Response · Authors · 2025-12-03
> > > > > **Response to Reviewer eRb5 (2/2)**
> > > > >
> > > > > 4.2. **Proof**
> > > > >
> > > > > $\text{FDR} = \sum_{(j,k) \in \mathcal{H}\_0} \mathbb{E}\left[ \frac{\mathbb{I}(H_{jk} \text{ is rejected})}{R \vee 1} \right]$
> > > > >
> > > > > Define the event $E_{jk} = \{p_{jk}^{\text{weighted}} \leq \frac{q \cdot R}{N}\}$. Then:
> > > > >
> > > > > $\text{FDR} = \sum_{(j,k) \in \mathcal{H}\_0} \mathbb{E}\left[ \frac{\mathbb{I}(E_{jk})}{R \vee 1} \right]$
> > > > >
> > > > >
> > > > > Let $\mathcal{F}\_{-jk}$ be the $\sigma$-algebra generated by all $p$-values and weights except $p_{jk}$. Define $R_{-jk}$ as the number of rejections obtained by applying the weighted BH procedure after excluding $H_{jk}$.
> > > > >
> > > > > From the properties of the BH procedure:
> > > > > * $R \geq R_{-jk}$
> > > > > * If $p_{jk}^{\text{weighted}} > \frac{q \cdot R_{-jk}}{N}$, then $R = R_{-jk}$.
> > > > >
> > > > > Thus:
> > > > >
> > > > > $\frac{\mathbb{I}(E_{jk})}{R \vee 1} \leq \frac{\mathbb{I}\left(p_{jk}^{\text{weighted}} \leq \frac{q \cdot R_{-jk}}{N}\right)}{R_{-jk} \vee 1}.$
> > > > >
> > > > >
> > > > >
> > > > > Take the conditional expectation:
> > > > >
> > > > >
> > > > > $\mathbb{E}\left[ \frac{\mathbb{I}(E_{jk})}{R \vee 1} \mid\mathcal{F}\_{-jk} \right] \leq \mathbb{E}\left[ \frac{\mathbb{I}\left(p_{jk}^{\text{weighted}} \leq \frac{q \cdot R_{-jk}}{N}\right)}{R_{-jk} \vee 1} \mid
> > > > > \mathcal{F}_{-jk} \right].$
> > > > >
> > > > > Given $\mathcal{F}\_{-jk}, R_{-jk}$ is fixed. The event $p_{jk}^{\text{weighted}} \leq \frac{q \cdot R_{-jk}}{N}$ is equivalent to $p_{jk} \leq \frac{q \cdot R_{-jk} \cdot w_{jk}}{N}$.
> > > > >
> > > > > By the conservativeness of the $p$-values (for $H_{jk} \in \mathcal{H}\_0)$:
> > > > > $\mathbb{P}\left(p_{jk} \leq \alpha \middle| \mathcal{F}_{-jk}\right) \leq \alpha \quad \text{for any } \alpha \in [0,1].$
> > > > >
> > > > > Set $\alpha = \frac{q \cdot R_{-jk} \cdot w_{jk}}{N}$, yielding:
> > > > > $\mathbb{E}\left[ \mathbb{I}\left(p_{jk}^{\text{weighted}} \leq \frac{q \cdot R_{-jk}}{N}\right) \middle| \mathcal{F}\_{-jk} \right] \leq \frac{q \cdot R_{-jk} \cdot w_{jk}}{N}.$
> > > > >
> > > > > Therefore:
> > > > > $\mathbb{E}\left[ \frac{\mathbb{I}\left(p_{jk}^{\text{weighted}} \leq \frac{q \cdot R_{-jk}}{N}\right)}{R_{-jk} \vee 1} \middle| \mathcal{F}\_{-jk} \right] \leq \frac{q \cdot w_{jk}}{N} \quad \text{when } R_{-jk} \geq 1 .$
> > > > > When $R_{-jk} = 0$, both sides are zero.
> > > > >
> > > > >
> > > > > Take the full expectation:
> > > > > $\mathbb{E}\left[ \frac{\mathbb{I}(E_{jk})}{R \vee 1} \right] \leq \frac{q \cdot w_{jk}}{N}.$
> > > > >
> > > > > Sum over all null hypotheses:
> > > > > $\text{FDR} \leq \sum_{(j,k) \in \mathcal{H}\_0} \frac{q \cdot w_{jk}}{N} = q \cdot \frac{\sum_{(j,k) \in \mathcal{H}\_0} w_{jk}}{N}.$
> > > > >
> > > > > From the weight condition $\sum_{(j,k) \in \mathcal{H}\_0} w_{jk} \leq \sum_{j,k} w_{jk} = N$,
> > > > >
> > > > > we obtain:
> > > > > $\text{FDR} \leq q \cdot \frac{\sum_{\mathcal{H}\_0} w_{jk}}{N} \leq q \cdot \frac{N}{N} = q.$
> > > > >
> > > > >
> > > > > 4.3. **Boundary Cases**
> > > > >
> > > > > * When $R = 0$, $FDR = 0$, satisfying the control condition.
> > > > > * When all weights are equal ($w_{jk} = 1$), the method reduces to the standard BH procedure.
> > > > >
> > > > >
> > > > >
> > > > > **5. Experimental results**
> > > > >
> > > > > If no weights are set, the proportion of samples in each interval of the selection set is consistent with the trend of the overall sample distribution. If needed, weights can be integrated to control the proportion of each interval in the selection set. We take the results of Task $6$ under Setting $3$ as an example in the table below.
> > > > >
> > > > >
> > > > >
> > > > > | *k* | | *No weights* |  | || *To focus on* | *a specific interval* || *To balance* | *across intervals* |
> > > > > |:-:|-|:-|:-|-|-|:-|:-|-|:-|:-|
> > > > > |  | | *All* | *Selected* | ||*All* | *Selected* ||*All* | *Selected* |
> > > > > | *1* | | ░ 5.9%| ▓▓ 13.1% | | |░ 5.9%| ▓▓ 11.8% | |░ 5.9%| ▓▓▓▓ 24.6% |
> > > > > | *2* | |░ 3.6%| ▓ 9.6% | | |░ 3.6%| ▓▓▓▓▓▓▓▓▓▓▓ 69.4% | |░ 3.6%| ▓▓▓▓ 25.4% |
> > > > > | *3* | |░ 6.4%| ▓ 9.4% | | |░ 6.4%| ▓ 4.7% | |░ 6.4%| ▓▓▓▓ 25.1% |
> > > > > | *4* | |░░░░░░░░ 42.8%| ▓▓▓▓▓▓▓▓▓▓ 68.0% | | |░░░░░░░░ 42.8%| ▓▓▓ 14.1%| |░░░░░░░░ 42.8% |▓▓▓▓ 24.9% |
> > > > >
> > > > >
> > > > >
> > > > >
> > > > >
> > > > >
> > > > >
> > > > > ---
> > > > >
> > > > >
> > > > > ### **2. Regarding the comparison with ACS**
> > > > >
> > > > > We appreciate the reviewer's engagement with the other reviewers' comments, though we wish to clarify several points. The core focus of ACS [1] is adaptation rather than multi-condition inference, which differs from the problem setting that MCCS aims to solve. In our work, we have developed a method with rigorous FDR guarantees and scalability across diverse scenarios for the specific challenge under study. While we acknowledge the foundational contributions of prior works such as ACS, our approach addresses a non-overlapping research gap. We do not dispute that ACS could potentially be extended to multivariate responses; indeed, such extensibility is one of its strengths. However, the current version of the corresponding paper explicitly defines  $Y \in \mathbb{R}$  in Section 2.1, and we believe that evaluating our contribution based on a not-yet-realized extension is not well-supported. More importantly, the absence of experimental validation in ACS for the problem considered in our work cannot be overlooked. As ICLR is widely recognized as a venue that emphasizes empirical rigor, we contend that our comprehensive experimental results warrant serious consideration.
> > > > >
> > > > > [1] Gui Y, et al. ACS: An interactive framework for conformal selection. arXiv:2507.15825, 2025.

---

### Official Review · Reviewer_yTUb · 2025-10-30

**Soundness:** 3
**Presentation:** 3
**Contribution:** 2
**Rating:** 4
**Confidence:** 4

**Summary:**

This paper studies the conformal selection (CS) problem, that is, to select --- among many test points --- those samples whose unknown responses satisfy a pre-specified property. While previous conformal selection method addresses a one-sided property, this paper considers multiple conditions such as an interval or the combination of two one-sided intervals. The authors first propose that taking the intersection/union of two conformal selection sets invalidates the FDR control. Then, for conjunctive conditions, they propose a method that designs a conformity score specialized for addressing such properties in the CS framework. For disjunctive conditions, they propose to select from all p-values for each test point and each property with BH, and show the FDR control. The proposed methods are demonstrated in extensive simulations and real-world applications.

**Strengths:**

1. The paper is tightly structured and clearly written.
2. The problem addressed is of practical relevance.
3. The discussion on technical challenges is precise and convincing.
4. The numerical experiments are extensive and solid.

**Weaknesses:**

1. The FDR guarantee for disjunctive conditions is hard to interpret (see my Q1 below).
2. The solution seems over-complicated (see my Q2 below). Seems a simple strategy can greatly simplify the proposal.

**Questions:**

1. What does the FDR control for selecting among all the $(j,k)$ pairs in Algorithm 3 mean? The definition is not explicitly given, and based on my guess, the practical interpretation is a bit weird to me. Ideally we want an FDR over *samples* so that at least $1-\alpha$ fraction of them satisfy any of the conditions. However, in the current setup, consider 2 conditions, and suppose a sample j is selected for both $(j,1)$ and $(j,2)$, its meaning would be a bit strange, and I'm not sure the FDR over all the pairs can reflect the practical need very well.
2. I was wondering if the problem can be solved via a simpler strategy. Suppose we want to select samples with $Y\in (-\infty, c_1] \cup [c_2,+\infty)$. Can we define a property as $Y^* = \mathbf{1}_{Y\in (-\infty, c_1] \cup [c_2,+\infty)}$ and directly apply the conformal selection method?

---

> ### Author Response · Authors · 2025-11-25
> **Response to Reviewer yTUb**
>
> Thanks for your recognition and the valuable suggestions. Please find our response below.
>
>
> ### **1. Explanation of the FDR [W1, Q1]**
>
>
> Thank you for allowing us to explain the FDR in our method. We will clarify the rationale and practical implications of this design, drawing on the theoretical foundations laid out in Appendix A.1 of the manuscript.
>
>
> In Algorithm 3, the FDR control is applied holistically to all $(j,k)$ pairs, where $j$ indexes test samples and $k$ indexes conditions. This means that the algorithm governs the proportion of false discoveries among all individual claims of the form "sample $j$ satisfies condition $k$."  The core theoretical insight from Appendix A.1 is that this granular control at the $(j,k)$ level naturally extends to guarantee the FDR for the overarching goal of selecting samples that satisfy any condition in the target set $I_{Target} = \cup_k I_k$. This is derived from a fundamental inclusion relationship: if a sample $j$ does not belong to the entire target region $I_{Target}$ (i.e., $y_{n+j} \notin I_{Target}$), then it necessarily does not belong to any individual sub-interval $I_k$ (i.e., $y_{n+j} \notin I_k$ for all $k$). Consequently, the set of false discoveries at the sample level (where samples are incorrectly selected relative to $I_{Target}$) is a subset of the false discoveries at the $(j,k)$ pair level. Mathematically, Appendix A.1 establishes the inequality:
>
> $FDR_{sample} = \mathbb{E}\left[ \frac{ |\\{ j \in R : y_{n+j} \notin I_{\text{Target}} \\}| }{ |R| \vee 1 } \right] \leq\mathbb{E}\left[ \frac{ |\\{ (j,k) \in \mathcal{S} : y_{n+j} \notin I_k \\}| }{ |\mathcal{S}| \vee 1 } \right] = FDR$,
>
> where FDR is controlled at level $q$ by Theorem 4.1 under exchangeability conditions. Thus, the sample-level FDR is also bounded by $q$, ensuring that the derived set $R = \\{ j \mid \exists k, (j,k) \in S \\}$ maintains FDR control approximately at $q$.
>
> From a practical standpoint, the output structure of Algorithm 3, which consists of the set $S$ of $(j,k)$ pairs, provides users with flexibility and transparency. Users can effortlessly derive the sample-level set $R$ by iterating through $S$ and collecting unique sample indices $j$, a computationally efficient process that requires no additional statistical processing. Moreover, the $(j,k)$ output allows users to query exactly which conditions each sample satisfies; in drug discovery, this might mean identifying that sample $1$ fulfills both condition $1$ (e.g., high binding affinity) and condition $2$ (e.g., low toxicity). This granularity is valuable for applications requiring detailed property claims.
>
> In summary, the FDR control over $(j,k)$ pairs in Algorithm 3 is a deliberate and robust design choice that ensures statistical rigor while offering practical flexibility. We added a reference to Appendix A.1 in Section 3.1 of the revised manuscript to clarify this design.
>
>
>
> ---
>
>
>
>
> ### 2. **On the necessity of the method design. [W2, Q2]**
>
> Thank you for this insightful concern. A method similar to $Y^*=\mathbb{1}_{Y\in(-\infty, c_1]\cup[c_2, +\infty)}$ has already been used as a baseline, which has been analyzed in Appendix B.2. We further provide a detailed explanation in the General Response.

---

### Official Review · Reviewer_GiWV · 2025-10-30

**Soundness:** 3
**Presentation:** 3
**Contribution:** 1
**Rating:** 4
**Confidence:** 4

**Summary:**

The paper considers the problem of selecting samples with FDR control, where multiple conjunctive or disjunctive intervals characterize the selection criterion. The proposed method generalizes *conformal selection* by Jin and Candès, and is evaluated in numerical experiments.

**Strengths:**

The writing of this paper is clear and easy to follow; the technical derivation is rigorous.

**Weaknesses:**

The problem raised in this paper seems to have already been solved by existing works in full generality.
For example, [1] considers the selection criterion characterized by a general set and allows it to
depend on the covariate, where the idea is to design the score reflecting the likelihood of $Y$
falling into the property set. The proposed (adaptive) method therein achieves
finite-sample FDR control, with cfBH as a special example. It would be helpful if the author
could clarify the contributions of the work.

**References**

1. Gui, Yu, et al. "ACS: An interactive framework for conformal selection." arXiv preprint arXiv:2507.15825 (2025).

**Questions:**

Refer to the "Weaknesses" section.

---

> ### Author Response · Authors · 2025-11-25
> **Response to Reviewer GiWV**
>
> Thanks for your recognition and the valuable suggestions. Please find our response below.
>
>
>
> ### **Clarification of Differences with ACS and Our Contribution [W1, Q1]**
>
> We appreciate the comment regarding the relationship between our work and the existing work of ACS [1]. The two works are complementary rather than overlapping, and the existence of ACS does not diminish MCCS's contributions. Below, we elaborate on this perspective:
>
> * **Motivation**: The core motivations of ACS and MCCS are distinct. ACS emphasized how the selection target varies with covariates, while MCCS focused on what structure constitutes the target by extending selection criteria to multi-condition forms.
> * **Scope**: ACS operated actually in single-condition settings, as evidenced by its use of intervals like $(-\infty, c(X_i)]$ or discrete sets (e.g., $\\{0\\}$), which remain single-condition by our definition. The original ACS text neither experimentally verified nor expressed support for multi-interval combinations, thereby reinforcing its single-condition focus. MCCS, however, explicitly defined and handled multi-condition scenarios with rigorous FDR control.
> * **Generality**: ACS is inherently limited to univariate responses, as it defined $Y_j$ as a real scalar, and extending it to multivariate settings would require new theoretical derivations that are not provided. MCCS, by design, supports multivariate responses and has been empirically verified. Furthermore, MCCS scales to extremely high-dimensional and multi-class tasks (Appendix D), demonstrating broader applicability beyond ACS's univariate constraints.
>
> In summary, MCCS's contribution lies in being the first to systematically solve a long-overlooked problem: while existing works, including ACS, default to single-condition targets (e.g., one-sided intervals), MCCS provides an approach for diverse multi-condition criteria with finite-sample FDR guarantees. This represents a foundational advance, complementing ACS's strengths in process optimization and underscoring MCCS's novelty in addressing criterion complexity. We added a summary of this point in Conclusion of the revised manuscript.
>
> [1] Gui Y, et al. ACS: An interactive framework for conformal selection. arXiv:2507.15825, 2025.

---

### Author Response · Authors · 2025-11-25
**General Response**

We sincerely appreciate the reviewers' thoughtful feedback and encouraging comments on our work. We are gratified that reviewers recognize the **practical relevance** of the problem addressed (*yTUb*) and that our work **fills an important gap** in conformal selection research (*2sng*). The theoretical contributions are highlighted as **rigorous** in technical derivation (*GiWV*), **sound** with non-trivial extensions (*eRb5*), and providing finite-sample FDR **guarantees** (*2sng*), while the discussion on technical challenges is noted as **precise and convincing** (*yTUb*). Methodologically, the **novel extension** to multi-condition settings (*2sng*) and the **broad scope** enabling handling of conjunctive, disjunctive, and multivariate scenarios (*2sng*) are emphasized. Experimentally, the **extensive and solid** evaluations (*yTUb*), **comprehensive** assessment across diverse modalities (*eRb5*), and **appropriate comparisons** that expose limitations of naive baselines (*eRb5*) are valued. Finally, the presentation is praised for being **clear and easy to follow** (*GiWV*), with a **tightly structured** and logical flow (*yTUb*), **clear exposition** (*eRb5*), and illustrative figures that aid in understanding core concepts (*eRb5*).



The reviews allow us to strengthen our manuscript, and the changes are summarized below:

* Added a reference to Appendix A.1 to clarify the design of FDR in Line 135-136. [*yTUb*]
* Added experiments on large $K$ in Line 404-410 and Line 411-420. [*eRb5*]
* Added experiments on analysis on interval-specific contribution in Line 404-410 and Line 421-427. [*eRb5*]
* Added experiments on other baselines in Appendix B.6. [*2sng*]
* Revised Conclusion to clarify our contribution in Line 529-531. [*GiWV*]
* Revised Conclusion to emphasize the necessity of the method design in Line 532-535. [*yTUb, eRb5*]
* Added experiments on efficiency optimization method in Line 504-523. [*2sng*]
* Added more experiments on $Ind$-like baselines in Appendix B.3. [*eRb5*]
* Added theoretical analysis and experiments on interval-specific preferences in Appendix B.4. [*eRb5*]

For clarity, we highlight the revised part of the manuscript in **blue** color. Here, we provide response to a common concern of two reviewers.


---

### **On the necessity of the method design**


We acknowledge reviewers' (*yTUb* and *eRb5*) concerns about the necessity of MCCS compared to simple alternatives that reduce multi-condition problems to single-condition ones. These alternatives align conceptually with the $Ind$ baseline detailed in Appendix B.2, which uses a scalar function to encode multi-condition logic. In the original manuscript, we present empirical results in Section 5.1, illustrating a critical drawback of $Ind$-like methods: it relies solely on a single $p$-value obtained after the condition transformation, which may make it more sensitive to errors in the process, leading to low power and overly conservative FDR. Here, we reproduce key results at the nominal FDR level of $0.3$.
The results in the table show that our MCCS can achieve higher power with more precise FDR control than $Ind$-like methods. Similar pheonomenon was demonstrated in prior work (e.g., mCS [2]), where dedicated algorithms can outperform indicator-based baselines in multivariate settings.



| Method | |  |  |FDR| | |  | | Power|| |
|:-:|:-:|:-:|:-:|:-:|:-:|:-:|:-:|:-:|:-:|:-:|:-:|
| |  |*Conjunctive-Univariate* | *Conjunctive-Multivariate*| *Disjunctive-Univariate*|*Disjunctive-Multivariate*| |*Conjunctive-Univariate* | *Conjunctive-Multivariate*| *Disjunctive-Univariate* |*Disjunctive-Multivariate*|
| *Ind* | | 0.2013 | 0.1491 |  0.2290  |  0.2172 | |  0.2126 | 0.1179 |  0.0000 | 0.0044 |
| *MCCS* | | 0.2874 |  0.2907 | 0.2848  | 0.2628 | | 0.9756 |  0.5348 | 0.9515 |  0.7013 |



In summary, while conceptual simplicity, simple methods proposed by reviewers equivalence to $Ind$ exposes inherent limitations in its conservative FDR control. In contrast, MCCS provides a robust solution that is fundamental for practical multi-condition scenarios. We added a summary of this point in Conclusion of the revised manuscript.


[1] Jin Y, et al. Selection by prediction with conformal p-values. JMLR2023.

[2] Tian B, et al. Multivariate Conformal Selection. ICML2025.

---

### Author Response · Authors · 2025-12-03
**Summary of Rebuttal for Area Chairs**

Dear Area Chair,


Thank you for your time in reviewing our rebuttal summary. The reviewers recognized our work **addresses an important gap in the area** (*2sng*), and described our theory as **rigorous and convincing** (*GiWV*,*yTUb*), and our experiments as **extensive and thorough** (*eRb5*,*yTUb*). Their primary concerns centered on the comparison with ACS [1] (*GiWV*,*eRb5*) and the demonstration of our method's advantages over the baseline method $Ind$ (*yTUb*, *eRb5*).

1. **Comparison with ACS**. The core of ACS lies in adaptation rather than multi-condition, which **differs from the main focus of our work**. We consider the two works to be complementary rather than overlapping.  Some reviewers suggested that ACS might inherently address the tasks examined in our manuscript, despite lacking relevant experiments. **We respectfully disagree with this view**, as the theoretical framework of ACS does not fully support the multi-condition tasks considered in our work. Moreover, the absence of empirical validation in ACS cannot be overlooked. Our work provides both theoretical and experimental contributions, which we believe align with the expectations of a conference such as ICLR that values empirical rigor alongside theoretical novelty.

2. **Superiority over $Ind$**. We have added comprehensive comparisons under diverse settings, including an evaluation of a variant suggested by the reviewers. **The results consistently confirm the superiority of our method.** Furthermore, we demonstrate that our approach can effectively regulate interval preferences, a capability that $Ind$-like methods fundamentally lack, **supported by rigorous theoretical analysis as requested by the reviewers**. These results strongly validate the advantages of our method.

In addition, we have provided reasonable experimental supplements and clarifications for all other points raised by the reviewers. Specifically, the main points are：


* **Definition of FDR** (*yTUb*): We have clarified the rationale behind our FDR definition by referring to the theoretical foundation in Appendix A.1.
* **Performance under larger $K$** (*eRb5*): Additional experiments were conducted with a value of $K$ one order of magnitude larger than in the original submission, confirming the robustness of our method under increased complexity.
* **Computational efficiency** (*2sng*): The efficiency of the BH procedure has been improved via a Quicksort[2]-based implementation, with experimental results showing notable acceleration, particularly for large $K$.

The key revisions have been incorporated into the revised manuscript, as summarized in our general response. **We believe that all reviewer concerns have been appropriately addressed.** Especially, we note that **reviewer *eRb5* indicated in the initial review that addressing the concerns would lead to a score increase**. We sincerely hope that the efforts reflected in our rebuttal and revised manuscript will be duly considered in the final decision.

Thank you once again for your attention and service.

Best, Authors



---


[1] Gui Y, et al. ACS: An interactive framework for conformal selection. arXiv:2507.15825, 2025.

[2] Hoare, C. A. R. (1961). Algorithm 65: FIND. Communications of the ACM, 4(7), 321-322.

---

### Meta-Review · Area_Chair_xRdM · 2026-01-03

**Summary:**

The paper proposes to address the problem of selecting high-quality candidates from large-scale datasets and proposes the Multi-Condition Conformal Selection algorithm with a nonconformity score. Four reviewers has engaged into the review process,three reviewers show negative rates(GiWV:4,yTUb:4,eRb5:4) and one reviewer shows positive rates(2sng:8). The summary of concerns of each reviewer is summarized as follows.
1) For reviewer GiWV, the reviewer poses 1 concern in the weakness part, i.e., lack of in-depth analysis with recent works(W1).
2) For reviewer yTUb, the reviewer poses 2 concerns in the weaknesses part and 2 questions such as hard to interpret FDR guarantee for disjunctive conditions(W1,Q1), over-complicated(W2,Q2).
3) For reviewer eRb5, the reviewer poses 2 concerns  in the weaknesses part and 3 questions such as Perceived Marginal Technical Contribution(W1,Q1), Clarity on Theoretical Novelty(W2),On Experimental Design and Interpretation(Q2) and On the Equivalent Interval Representations(Q3).
4) For reviewer 2sng, the reviewer poses 1 concern in the weaknesses part and 3 questions such as ;ack of in-depth analysis in experiments(W1), lack of computational requirement analysis(Q1), lack of  future discussion(Q2) and  concerns of efficiency on Global BH procedure(Q3).

For soundness, all reviewers agree that the proposed method is sound. For concerns raised by the reviewers, the authors have made point by point rebutals. The paper is somewhat sound with theoretical analysis and performance improvement. Therefore, the paper is accepted.

**Reviewer Concerns:**

Most of the concerns are addressed point by point from the authors. For eRb5, Q1,W1 are still outstanding to clear the major differences.

**Reviewer Scores:**

yTUb may improve its score.

---

### Decision · Program_Chairs · 2026-01-26

Accept (Poster)